# Learning threshold neurons via the "edge of stability"

**Kwangjun Ahn**
MIT EECS
Cambridge, MA
kjahn@mit.edu

**Sébastien Bubeck**
Microsoft Research
Redmond, WA
sebubeck@microsoft.com

**Sinho Chewi**
Institute for Advanced Study
Princeton, NJ
schewi@ias.edu

**Yin Tat Lee**
Microsoft Research
Redmond, WA
yintat@uw.edu

**Felipe Suarez**
Carnegie Mellon University
Pittsburgh, PA
felipesc@mit.edu

**Yi Zhang**
Microsoft Research
Redmond, WA
zhayi@microsoft.com

## Abstract

Existing analyses of neural network training often operate under the unrealistic assumption of an extremely small learning rate. This lies in stark contrast to practical wisdom and empirical studies, such as the work of J. Cohen et al. (ICLR 2021), which exhibit startling new phenomena (the "edge of stability" or "unstable convergence") and potential benefits for generalization in the large learning rate regime. Despite a flurry of recent works on this topic, however, the latter effect is still poorly understood. In this paper, we take a step towards understanding genuinely non-convex training dynamics with large learning rates by performing a detailed analysis of gradient descent for simplified models of two-layer neural networks. For these models, we provably establish the edge of stability phenomenon and discover a sharp phase transition for the step size below which the neural network fails to learn "threshold-like" neurons (i.e., neurons with a non-zero first-layer bias). This elucidates one possible mechanism by which the edge of stability can in fact lead to better generalization, as threshold neurons are basic building blocks with useful inductive bias for many tasks.

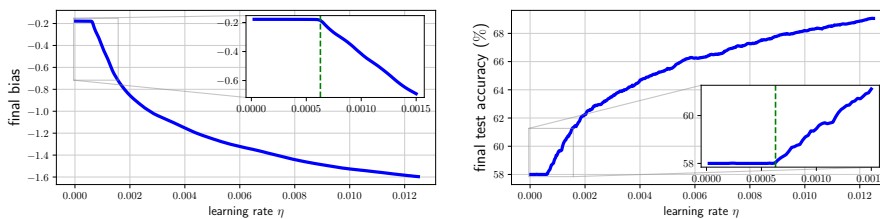

Figure 1: *Large step sizes are necessary to learn the "threshold neuron" of a ReLU network* (2) *for a simple binary classification task* (1)*.* We choose $d = 200$, $n = 300$, $\lambda = 3$, and run gradient descent with the logistic loss. The weights are initialized as $a^-, a^+ \sim \mathcal{N}(0, 1/(2d))$ and $b = 0$. For each learning rate $\eta$, we set the iteration number such that the total time elapsed (iteration $\times \eta$) is 10. The vertical dashed lines indicate our theoretical prediction of the phase transition phenomenon (precise threshold at $\eta = 8\pi/d^2$).

## 1 Introduction

How much do we understand about the training dynamics of neural networks? We begin with a simple and canonical learning task which indicates that the answer is still "far too little".

37th Conference on Neural Information Processing Systems (NeurIPS 2023).

*Motivating example:* Consider a binary classification task of labeled pairs $(\boldsymbol{x}^{(i)}, y^{(i)}) \in \mathbb{R}^d \times \{\pm 1\}$ where each covariate $\boldsymbol{x}^{(i)}$ consists of a 1-sparse vector (in an unknown basis) corrupted by additive Gaussian noise, and the label $y^{(i)}$ is the sign of the non-zero coordinate of the 1-sparse vector. Due to rotational symmetry, we can take the unknown basis to be the standard one and write

$$\boldsymbol{x}^{(i)} = \lambda y^{(i)} \boldsymbol{e}_{j(i)} + \boldsymbol{\xi}^{(i)} \in \mathbb{R}^d, \tag{1}$$

where $y^{(i)} \in \{\pm 1\}$ is a random label, $j(i) \in [d]$ is a random index, $\boldsymbol{\xi}^{(i)}$ is Gaussian noise, and $\lambda > 1$ is the unknown signal strength. In fact, (1) is a special case of the well-studied sparse coding model (Olshausen and Field, 1997; Vinje and Gallant, 2000; Olshausen and Field, 2004; Yang et al., 2009; Koehler and Risteski, 2018; Allen-Zhu and Li, 2022). We ask the following fundamental question:

*How do neural networks learn to solve the sparse coding problem* (1)*?*

In spite of the simplicity of the setting, a full resolution to this question requires a thorough understanding of surprisingly rich dynamics which ***lies out of reach of existing theory***. To illustrate this point, consider an extreme simplification in which the basis $\boldsymbol{e}_1, \ldots, \boldsymbol{e}_d$ is known in advance, for which it is natural to parametrize a two-layer ReLU network as

$$f(\boldsymbol{x}; a^-, a^+, b) = a^- \sum_{i=1}^d \mathrm{ReLU}\left(-\boldsymbol{x}[i] + b\right) + a^+ \sum_{i=1}^d \mathrm{ReLU}\left(+\boldsymbol{x}[i] + b\right). \tag{2}$$

The parametrization (2) respects the latent data structure (1) well: a good network has a negative bias $b$ to threshold out the noise, and has $a^- < 0$ and $a^+ > 0$ to output correct labels. We are particularly interested in understanding the mechanism by which the bias $b$ becomes negative, thereby allowing the non-linear ReLU activation to act as a threshold function; we refer to this as the problem of learning "threshold neurons". More broadly, such threshold neurons are of interest as they constitute basic building blocks for producing neural networks with useful inductive bias.

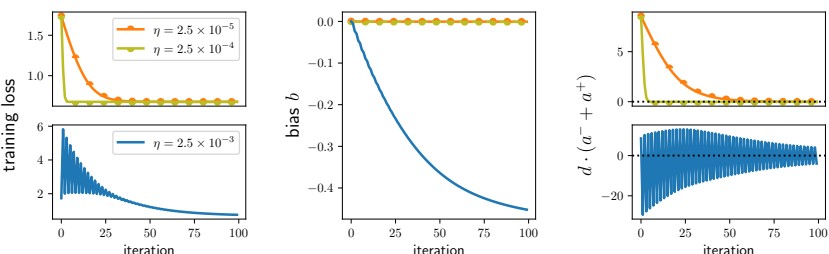

Figure 2: *Large learning rates lead to unexpected phenomena: non-monotonic loss and wild oscillations of weights.* We choose the same setting as Figure 1. With a small learning rate ($\eta = 2.5 \cdot 10^{-5}$), the bias does not decrease noticeably, and the same is true even when we increase the learning rate by ten times ($\eta = 2.5 \cdot 10^{-4}$). When we increase the learning rate by another ten times ($\eta = 2.5 \cdot 10^{-3}$), we finally see a noticeable decrease in the bias, but with this we observe unexpected behavior: *the loss decreases non-monotonically and the sum of second-layer weights* $d \cdot (a^- + a^+)$ *oscillates wildly.*

We train the parameters $a^-$, $a^+$, $b$ using gradient descent with step size $\eta > 0$ on the logistic loss $\sum_{i=1}^n \ell_{\mathrm{logi}}(y^{(i)} f(\boldsymbol{x}^{(i)}; a^-, a^+, b))$, where $\ell_{\mathrm{logi}}(z) := \log(1 + \exp(-z))$, and we report the results in Figures 1 and 2. The experiments reveal a compelling picture of the optimization dynamics.

- ■ *Large learning rates are necessary, both for generalization and for learning threshold neurons.* Figure 1 shows that the bias decreases and the test accuracy increases as we increase $\eta$; note that we plot the results after a fixed *time* (iteration $\times \eta$), so the observed results are not simply because larger learning rates track the continuous-time gradient flow for a longer time.
- ■ *Large learning rates lead to unexpected phenomena: non-monotonic loss and wild oscillations of* $a^- + a^+$. Figure 2 shows that large learning rates also induce stark phenomena, such as non-monotonic loss and large weight fluctuations, which lie firmly outside the explanatory power of existing analytic techniques based on principles from convex optimization.
- ■ *There is a phase transition between small and large learning rates.* In Figure 1, we zoom in on learning rates around $\eta \approx 0.0006$ and observe *sharp* phase transition phenomena.

We have presented these observations in the context of the simple ReLU network (2), but we emphasize that ***these findings are indicative of behaviors observed in practical neural network training settings.*** In Figure 3, we display results for a two-layer ReLU network trained on the full sparse coding model (1) with unknown basis, as well as a deep neural network trained on CIFAR-10. In each case, we again observe non-monotonic loss coupled with steadily decreasing bias parameters. For these richer models, the transition from small to large learning rates is oddly reminiscent of well-known separations between the "lazy training" or "NTK" regime Jacot et al. (2018) and the more expressive "feature learning" regime. For further experimental results, see Appendix B.

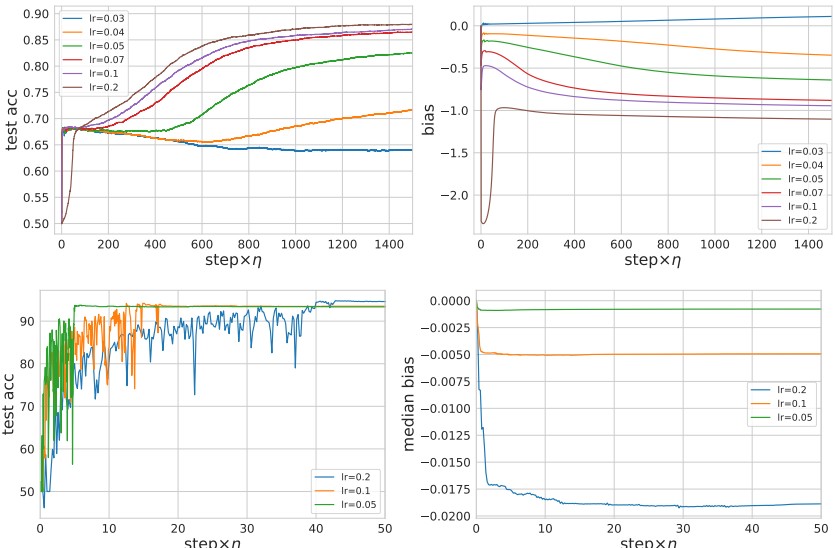

Figure 3: (Top) Results for training an over-parametrized two-layer neural network $f(\boldsymbol{x}; \boldsymbol{a}, \boldsymbol{W}, b) = \sum_{i=1}^{m} a_i \operatorname{ReLU}(\boldsymbol{w}_i^\top \boldsymbol{x} + b)$ with $m \gg d$ for the ***full sparse coding model*** (1)*;* in this setting, the basis vectors are unknown, and the neural network learn them through additional parameters $\boldsymbol{W} = (\boldsymbol{w}_i)_{i=1}^{m}$. Also, we use $m$ different weights $\boldsymbol{a} = (a_i)_{i=1}^{m}$ for the second layer. (Bottom) Full-batch gradient descent dynamics of ***ResNet-18 on (binary) CIFAR-10 with various learning rates.*** Details are deferred to Appendix B.

We currently do not have right tools to understand these phenomena. First of all, a drastic change in behavior between the small and the large learning rates cannot be captured through well-studied regimes, such as the "neural tangent kernel" (NTK) regime (Jacot et al., 2018; Allen-Zhu et al., 2019; Arora et al., 2019; Chizat et al., 2019; Du et al., 2019; Oymak and Soltanolkotabi, 2020) or the mean-field regime Chizat and Bach (2018); Mei et al. (2019); Chizat (2022); Nitanda et al. (2022); Rotskoff and Vanden-Eijnden (2022). In addition, understanding why a large learning rate is required to learn the bias is beyond the scope of prior theoretical works on the sparse coding model (Arora et al., 2015; Karp et al., 2021). Our inability to explain these findings points to a serious gap in our grasp of neural network training dynamics and calls for a detailed theoretical study.

## 1.1 Main scope of this work

In this work, we do not aim to understand the sparse coding problem (1) in its full generality. Instead, we pursue the more modest goal of shedding light on the following question.

> **Q.** What is the role of a large step size in learning the bias for the ReLU network (2)?

As discussed above, the dynamics of the simple ReLU network (2) is a microcosm of emergent phenomena beyond the convex optimization regime. In fact, there is a recent growing body of work (Cohen et al., 2021; Arora et al., 2022; Ahn et al., 2022; Lyu et al., 2022; Ma et al., 2022; Wang et al., 2022b; Chen and Bruna, 2023; Damian et al., 2023; Zhu et al., 2023) on training with large learning rates, which largely aims at explaining a striking empirical observation called the "***edge of stability (EoS)***" phenomenon.

The edge of stability (EoS) phenomenon is a set of distinctive behaviors observed recently by Cohen et al. (2021) when training neural networks with gradient descent (GD). Here we briefly summarize the salient features of the EoS and defer a discussion of prior work to Subsection 1.3. Recall that if we use GD to optimize an $L$-smooth loss function with step size $\eta$, then the well-known descent lemma from convex optimization ensures monotonic decrease in the loss so long as $L < 2/\eta$. In contrast, when $L > 2/\eta$, it is easy to see on simple convex quadratic examples that GD can be unstable (or divergent). The main observation of Cohen et al. (2021) is that when training neural networks[1] with constant step size $\eta > 0$, the largest eigenvalue of the Hessian at the current iterate (dubbed the "sharpness") initially increases during training ("progressive sharpening") and saturates near or above $2/\eta$ ("EoS").

A surprising message of the present work is that ***the answer to our main question is intimately related to the EoS***. Indeed, Figure 4 shows that the GD iterates of our motivating example exhibit the EoS during the initial phase of training when the bias decreases rapidly.

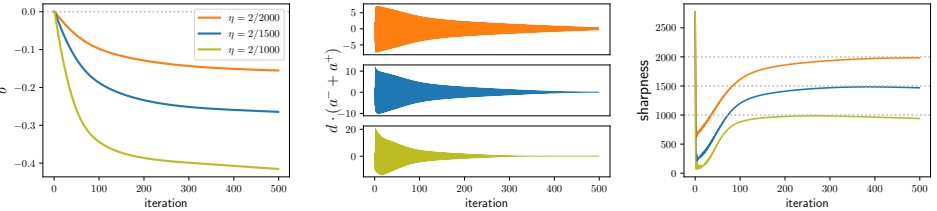

Figure 4: **Understanding our main question is surprisingly related to the EoS.** Under the same setting as Figure 1, we report the largest eigenvalue of the Hessian ("sharpness"), and observe that GD iterates lie in the EoS during the initial phase of training when there is a fast drop in the bias.

Consequently, we first set out to thoroughly understand the workings of the EoS phenomena through a simple example. Specifically, we consider a single-neuron linear neural network in dimension 1, corresponding to the loss

$$\mathbb{R}^2 \ni (x, y) \mapsto \ell(xy), \qquad \text{where } \ell \text{ is convex, even, and Lipschitz} . \tag{3}$$

Although toy models have appeared in works on the EoS (see Subsection 1.3), our example is simpler than all prior models, and we provably establish the EoS for (3) with transparent proofs.

We then use the newfound insights gleaned from the analysis of (3) to answer our main question. To the best of our knowledge, we provide the first explanation of the mechanism by which a large learning rate can be *necessary* for learning threshold neurons.

## 1.2 Our contributions

***Explaining the EoS with a single-neuron example.*** Although the EoS has been studied in various settings (see Subsection 1.3 for a discussion), these works either do not rigorously establish the EoS phenomenon, or they operate under complex settings with opaque assumptions. Here, we study a simple two-dimensional loss function, $(x, y) \mapsto \ell(xy)$, where $\ell$ is convex, even, and Lipschitz. Some examples include[2] $\ell(s) = \frac{1}{2}\log(1 + \exp(-s)) + \frac{1}{2}\log(1 + \exp(+s))$ and $\ell(s) = \sqrt{1 + s^2}$. Surprisingly, GD on this loss already exhibits rich behavior (Figure 5).

En route to this result, we rigorously establish the quasi-static dynamics formulated in Ma et al. (2022).

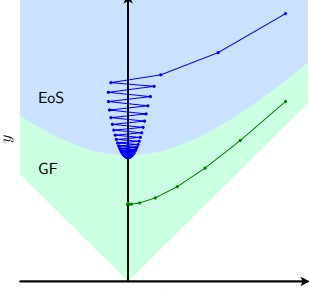

Figure 5: Illustration of two different regimes (the "gradient flow" regime and the "EoS" regime) of the GD dynamics.

---

[1]The phenomenon in Cohen et al. (2021) is most clearly observed for $\tanh$ activations, although the appendix of Cohen et al. (2021) contains thorough experimental results for various neural network architectures.

[2]Suppose that we have a single-layer linear neural network $f(x; a, b) = abx$, and that the data is drawn according to $x = 1$, $y \sim \mathsf{unif}(\{\pm 1\})$. Then, the population loss under the logistic loss is $(a, b) \mapsto \ell_{\mathrm{sym}}(ab)$ with $\ell_{\mathrm{sym}}(s) = \frac{1}{2}\log(1 + \exp(-s)) + \frac{1}{2}\log(1 + \exp(+s))$.

The elementary nature of our example leads to transparent arguments, and consequently our analysis isolates generalizable principles for "bouncing" dynamics. To demonstrate this, we use our insights to study our main question of learning threshold neurons.

***Learning threshold neurons with the mean model.*** The connection between the single-neuron example and the ReLU network (2) can already be anticipated via a comparison of the dynamics: *(i)* for the single neuron example, $x$ oscillates wildly while $y$ decreases (Figure 5); *(ii)* for the ReLU network (2), the sum of weights $(a^- + a^+)$ oscillates while $b$ decreases (Figure 2).

We study this example in Section 2 and delineate a transition from the "gradient flow" regime to the "EoS regime", depending on the step size $\eta$ and the initialization. Moreover, in the EoS regime, we rigorously establish asymptotics for the limiting sharpness which depend on the higher-order behavior of $\ell$. In particular, for the two losses mentioned above, the limiting sharpness is $2/\eta + O(\eta)$, whereas for losses $\ell$ which are exactly quadratic near the origin the limiting sharpness is $2/\eta + O(1)$.

In fact, this connection can be made formal by considering an approximation for the GD dynamics for the ReLU network (2). It turns out (see Subsection 3.1 for details) that during the initial phase of training, the dynamics of $A_t := d\left(a_t^- + a_t^+\right)$ and $b_t$ due to the ReLU network are well-approximated by the "rescaled" GD dynamics on the loss $(A, b) \mapsto \ell_{\text{sym}}(A \times g(b))$, where the step size for the $A$-dynamics is multiplied by $2d^2$, $g(b) := \mathbb{E}_{z \sim \mathcal{N}(0,1)} \text{ReLU}(z + b)$ is the "smoothed" ReLU, and $\ell_{\text{sym}}$ is the symmetrized logistic loss; see Subsection 3.1 and Figure 8. We refer to these dynamics as the ***mean model***. The mean model bears a great resemblance to the single-neuron example $(x, y) \mapsto \ell(xy)$, and hence we can leverage the techniques developed for the latter in order to study the former.

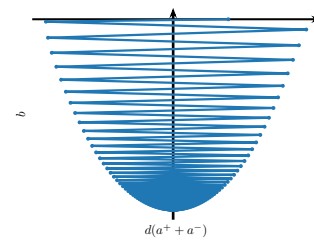

Figure 6: Illustration of GD dynamics on the ReLU network (2). The sum of weights $(a^- + a^+)$ oscillates while $b$ decreases.

Our main result for the mean model precisely explains the phase transition in Figure 1. For any $\delta > 0$,

- if $\eta \leq (8 - \delta)\pi/d^2$, then ***the mean model fails to learn threshold neurons***: the limiting bias satisfies $|b_\infty| = O_\delta(1/d^2)$.
- if $\eta \geq (8 + \delta)\pi/d^2$, then ***the mean model enters the EoS and learns threshold neurons***: the limiting bias satisfies $b_\infty \leq -\Omega_\delta(1)$.

## 1.3 Related work

***Edge of stability.*** Our work is motivated by the extensive empirical study of Cohen et al. (2021), which identified the EoS phenomenon. Subsequently, there has been a flurry of works aiming at developing a theoretical understanding of the EoS, which we briefly summarize here.

*Properties of the loss landscape.* The works (Ahn et al., 2022; Ma et al., 2022) study the properties of the loss landscape that lead to the EoS. Namely, Ahn et al. (2022) argue that the existence of forward-invariant subsets near the minimizers allows GD to convergence even in the unstable regime. They also explore various characteristics of EoS in terms of loss and iterates. Also, Ma et al. (2022) empirically show that the loss landscape of neural networks exhibits subquadratic growth locally around the minimizers. They prove that for a one-dimensional loss, subquadratic growth implies that GD finds a 2-periodic trajectory.

*Limiting dynamics.* Other works characterize the limiting dynamics of the EoS in various regimes. (Arora et al., 2022; Lyu et al., 2022) show that (normalized) GD tracks a "sharpness reduction flow" near the manifold of minimizers. The recent work of Damian et al. (2023) obtains a different predicted dynamics based on self-stabilization of the GD trajectory. Also, Ma et al. (2022) describes a quasi-static heuristic for the overall trajectory of GD when one component of the iterate is oscillating.

*Simple models and beyond.* Closely related to our own approach, there are prior works which carefully study simple models. Chen and Bruna (2023) prove global convergence of GD for the two-dimensional function $(x, y) \mapsto (xy - 1)^2$ and a single-neuron student-teacher setting; note that unlike our results, they do not study the limiting sharpness. Wang et al. (2022b) study progressive sharpening for a neural network model. Also, the recent and concurrent work of Zhu et al. (2023) studies the two-dimensional loss $(x, y) \mapsto (x^2y^2 - 1)^2$; to our knowledge, their work is the first to asymptotically and rigorously show that the limiting sharpness of GD is $2/\eta$ in a simple setting,

at least when initialized locally. In comparison, in Section 2, we perform a global analysis of the limiting sharpness of GD for $(x, y) \mapsto \ell(xy)$ for a class of convex, even, and Lipschitz losses $\ell$, and in doing so we clearly delineate the "gradient flow regime" from the "EoS regime".

***Effect of learning rate on learning.*** Recently, several works have sought to understand how the choice of learning rate affects the learning process, in terms of the properties of the resulting minima (Jastrzebski et al., 2018; Wu et al., 2018; Mulayoff et al., 2021; Nacson et al., 2022) and the behavior of optimization dynamics (Xing et al., 2018; Jastrzebski et al., 2019, 2020; Lewkowycz et al., 2020; Jastrzebski et al., 2021).

Li et al. (2019) demonstrate for a synthethic data distribution and a two-layer ReLU network model that choosing a larger step size for SGD helps with generalization. Subsequent works have shown similar phenomena for regression (Nakkiran, 2020; Wu et al., 2021; Ba et al., 2022), kernel ridge regression Beugnot et al. (2022), and linear diagonal networks Nacson et al. (2022). However, the large step sizes considered in these work still fall under the scope of descent lemma, and most prior works do not theoretically investigate the effect of large step size in the EoS regime. A notable exception is the work of Wang et al. (2022a), which studies the impact of learning rates greater than 2/smoothness for a matrix factorization problem. Also, the recent work of Andriushchenko et al. (2023) seeks to explain the generalization benefit of SGD in the large step size regime by relying on a heuristic SDE model for the case of linear diagonal networks. Despite this similarity, their main scope is quite different from ours, as we *(i)* focus on GD instead of SGD and *(ii)* establish a direct and detailed analysis of the GD dynamics for a model of the motivating sparse coding example.

## 2  Single-neuron linear network

In this section, we analyze the single-neuron linear network model $(x, y) \mapsto f(x, y) \coloneqq \ell(x \times y)$.

### 2.1  Basic properties and assumptions

***Basic properties.*** If $\ell$ is minimized at $0$, then the *global minimizers* of $f$ are the $x$- and $y$-axes. The GD *iterates* $x_t, y_t$, for step size $\eta > 0$ and iteration $t \geq 0$ can be written as

$$x_{t+1} = x_t - \eta \, \ell'(x_t y_t) \, y_t \,, \qquad y_{t+1} = y_t - \eta \, \ell'(x_t y_t) \, x_t \,.$$

***Assumptions.*** From here onward, we assume $\eta < 1$ and the following conditions on $\ell : \mathbb{R} \to \mathbb{R}$.

(A1) $\ell$ is convex, even, 1-Lipschitz, and of class $\mathcal{C}^2$ near the origin with $\ell''(0) = 1$.
(A2) There exist constants $\beta > 1$ and $c > 0$ with the following property: for all $s \neq 0$,

$$\ell'(s)/s \leq 1 - c \, |s|^\beta \, \mathbb{1}\{|s| \leq c\} \,.$$

We allow $\beta = +\infty$, in which case we simply require that $\frac{\ell'(s)}{s} \leq 1$ for all $s \neq 0$.

Assumption (A2) imposes decay of $s \mapsto \ell'(s)/s$ locally away from the origin in order to obtain more fine-grained results on the limiting sharpness in Theorem 2. As we show in Lemma 5 below, when $\ell$ is smooth and has a strictly negative fourth derivative at the origin, then Assumption (A2) holds with $\beta = 2$. See Example 1 for some simple examples of losses satisfying our assumptions.

### 2.2  Two different regimes for GD depending on the step size

Before stating rigorous results, in this section we begin by giving an intuitive understanding of the GD dynamics. It turns out that for a given initialization $(x_0, y_0)$, there are two different regimes for the GD dynamics depending on the step size $\eta$. Namely, there exists a threshold on the step size such that *(i)* below the threshold, GD remains close to the gradient flow for all time, and *(ii)* above the threshold, GD enters the edge of stability and diverges away from the gradient flow. See Figure 9.

First, recall that the GD dynamics are symmetric in $x, y$ and that the lines $y = \pm x$ are invariant. Hence, we may assume without loss of generality that

$$y_0 > x_0 > 0 \,, \quad y_t > |x_t| \text{ for all } t \geq 1 \,, \quad \text{and GD converges to } (0, y_\infty) \text{ for } y_\infty > 0 \,.$$

From the expression (8) for the Hessian of $f$ and our normalization $\ell''(0) = 1$, it follows that the sharpness (the largest eigenvalue of loss Hessian) reached by GD in this example is precisely $y_\infty^2$.

Initially, in both regimes, the GD dynamics tracks the continuous-time gradient flow. Our first observation is that the gradient flow admits a conserved quantity, thereby allowing us to predict the dynamics in this initial phase.

**Lemma 1** (conserved quantity). *Along the gradient flow for $f$, the quantity $y^2 - x^2$ is conserved.*

*Proof.* Differentiating $y_t^2 - x_t^2$ with respect to $t$ gives $2y_t \left(-\ell'(x_t y_t) x_t\right) - 2x_t \left(-\ell(x_t y_t) y_t\right) = 0$. □

Lemma 1 implies that the gradient flow converges to $(0, y_\infty^{\mathrm{GF}}) = (0, \sqrt{y_0^2 - x_0^2})$. For GD with step size $\eta > 0$, the quantity $y^2 - x^2$ is no longer conserved, but we show in Lemma 6 that it is *approximately* conserved until the GD iterate lies close to the $y$-axis. Hence, GD initialized at $(x_0, y_0)$ also reaches the $y$-axis approximately at the point $(x_{t_0}, y_{t_0}) \approx (0, \sqrt{y_0^2 - x_0^2})$.

At this point, GD either approximately converges to the gradient flow solution $(0, \sqrt{y_0^2 - x_0^2})$ or diverges away from it, depending on whether or not $y_{t_0}^2 > 2/\eta$. To see this, for $|x_{t_0} y_{t_0}| \ll 1$, we can Taylor expand $\ell'$ near zero to obtain the approximate dynamics for $x$ (recalling $\ell''(0) = 1$),

$$x_{t_0+1} \approx x_{t_0} - \eta x_{t_0} y_{t_0}^2 = (1 - \eta y_{t_0}^2) x_{t_0}. \tag{4}$$

From (4), we deduce the following conclusions.

(i) If $y_{t_0}^2 < 2/\eta$, then $|1 - \eta y_{t_0}^2| < 1$. Since $y_t$ is decreasing, it implies that $|1 - \eta y_t^2| < 1$ for all $t \geq t_0$, and so $|x_t|$ converges to zero exponentially fast.

(ii) On the other hand, if $y_{t_0}^2 > 2/\eta$, then $|1 - \eta y_{t_0}^2| > 1$, i.e., the magnitude of $x_{t_0}$ increases in the next iteration, and hence GD cannot stabilize. In fact, in the approximate dynamics, $x_{t_0+1}$ has the opposite sign as $x_{t_0}$, i.e., $x_{t_0}$ jumps across the $y$-axis. One can show that the "bouncing" of the $x$ variable continues until $y_t^2$ has decreased past $2/\eta$, at which point we are in the previous case and GD approximately converges to $(0, 2/\eta)$.

This reasoning, combined with the expression for the Hessian of $f$, shows that

$$\mathsf{sharpness}(0, y_\infty) := \lambda_{\max}\left(\nabla^2 f(0, y_\infty)\right) \approx \min\{y_0^2 - x_0^2, \, 2/\eta\}$$
$$= \min\{\text{gradient flow sharpness}, \text{ EoS prediction}\}.$$

Accordingly, we refer to the case $y_0^2 - x_0^2 < 2/\eta$ as the **gradient flow regime**, and the case $y_0^2 - x_0^2 > 2/\eta$ as the **EoS regime**.

See Figure 5 and Figure 9 for illustrations of these two regimes; see also Figure 10 for detailed illustrations of the EoS regime. In the subsequent sections, we aim to make the above reasoning rigorous. For example, instead of the approximate dynamics (4), we consider the original GD dynamics and justify the Taylor approximation. Also, in the EoS regime, rather than loosely asserting that $|x_t| \searrow 0$ exponentially fast and hence the dynamics stabilizes "quickly" once $y_t^2 < 2/\eta$, we track precisely how long this convergence takes so that we can bound the gap between the limiting sharpness and the prediction $2/\eta$.

## 2.3 Results

**Gradient flow regime.** Our first rigorous result is that when $y_0^2 - x_0^2 = (2-\delta)/\eta$ for some constant $\delta \in (0, 2)$, then the limiting sharpness of GD with step size $\eta$ is $y_0^2 - x_0^2 + O(1) = (2-\delta)/\eta + O(1)$, which is precisely the sharpness attained by the gradient flow up to a controlled error term.

In fact, our theorem is slightly more general, as it covers initializations in which $\delta$ can scale mildly with $\eta$. The precise statement is as follows.

**Theorem 1** (gradient flow regime; see Subsection C.2). *Suppose we run GD with step size $\eta > 0$ on the objective $f$, where $f(x, y) := \ell(xy)$, and $\ell$ satisfies Assumptions (A1) and (A2). Let $(\tilde{x}, \tilde{y}) \in \mathbb{R}^2$ satisfy $\tilde{y} > \tilde{x} > 0$ with $\tilde{y}^2 - \tilde{x}^2 = 1$. Suppose we initialize GD at $(x_0, y_0) := (\frac{2-\delta}{\eta})^{1/2} (\tilde{x}, \tilde{y})$, where $\delta \in (0, 2)$ and $\eta \lesssim \delta^{1/2} \wedge (2 - \delta)$. Then, GD converges to $(0, y_\infty)$ satisfying*

$$\frac{2-\delta}{\eta} - O(2-\delta) - O\left(\frac{\eta}{\min\{\delta, 2-\delta\}}\right) \leq \lambda_{\max}\left(\nabla^2 f(0, y_\infty)\right) \leq \frac{2-\delta}{\eta} + O\left(\frac{\eta}{2-\delta}\right),$$

*where the implied constants depend on $\tilde{x}$, $\tilde{y}$, and $\ell$, but not on $\delta$, $\eta$.*

The proof of Theorem 1 is based on a two-stage analysis. In the first stage, we use Lemma 6 on the approximate conservation of $y^2 - x^2$ along GD in order to show that GD lands near the $y$-axis with $y_{t_0}^2 \approx {}^{(2-\delta)}/\eta$. In the second stage, we use the assumptions on $\ell$ in order to control the rate of convergence of $|x_t|$ to 0, which is subsequently used to control the final deviation of $y_\infty^2$ from ${}^{(2-\delta)}/\eta$.

***EoS regime.*** Our next result states that when $y_0^2 - x_0^2 > 2/\eta$, then the limiting sharpness of GD is close to the EoS prediction of $2/\eta$, up to an error term which depends on the exponent $\beta$ in (A2).

**Theorem 2** (EoS; see Subsection C.4). *Suppose we run GD on $f$ with step size $\eta > 0$, where $f(x, y) := \ell(xy)$, and $\ell$ satisfies (A1) and (A2). Let $(\tilde{x}, \tilde{y}) \in \mathbb{R}^2$ satisfy $\tilde{y} > \tilde{x} > 0$ with $\tilde{y}^2 - \tilde{x}^2 = 1$. Suppose we initialize GD at $(x_0, y_0) := \sqrt{{}^{(2+\delta)}/\eta}\,(\tilde{x}, \tilde{y})$, where $\delta > 0$ is a constant. Also, assume that for all $t \geq 1$ such that $y_t^2 > 2/\eta$, we have $x_t \neq 0$. Then, GD converges to $(0, y_\infty)$ satisfying*

$$2/\eta - O(\eta^{1/(\beta-1)}) \leq \lambda_{\max}\big(\nabla^2 f(0, y_\infty)\big) \leq 2/\eta\,,$$

*where the implied constants depend on $\tilde{x}$, $\tilde{y}$, $\delta \wedge 1$, and $\ell$, but not on $\eta$.*

***Remarks on the assumptions.*** The initialization in our results is such that both $y_0$ and $y_0 - x_0$ are on the same scale, i.e., $y_0, y_0 - x_0 = \Theta(1/\sqrt{\eta})$. This rules out extreme initializations such as $y_0 \approx x_0$, which are problematic because they lie too close to the invariant line $y = x$. Since our aim in this work is not to explore every edge case, we focus on this setting for simplicity. Moreover, we imposed the assumption that the iterates of GD do not exactly hit the $y$-axis before crossing $y^2 = 2/\eta$. This is necessary because if $x_t = 0$ for some iteration $t$, then $(x_{t'}, y_{t'}) = (x_t, y_t)$ for all $t' > t$, and hence the limiting sharpness may not be close to $2/\eta$. This assumption holds generically, e.g., if we perturb each iterate of GD with a vanishing amount of noise from a continuous distribution, and we conjecture that for any $\eta > 0$, the assumption holds for all but a measure zero set of initializations.

When $\beta = +\infty$, which is the case for the Huber loss in Example 1, the limiting sharpness is $2/\eta + O(1)$. When $\beta = 2$, which is the case for the logistic and square root losses in Example 1, the limiting sharpness is $2/\eta + O(\eta)$. Numerical experiments show that ***our error bound of $O(\eta^{1/(\beta-1)})$ is sharp***; see Figure 11 below.

We make a few remark about the proof. As we outline the proof in Subsection C.3, in turns out in order to bound the gap $2/\eta - y_\infty^2$, the proof requires a control of the size $|x_{\mathbf{t}} y_{\mathbf{t}}|$, where $\mathbf{t}$ is the first iteration such that $y_{\mathbf{t}}^2$ crosses $2/\eta$. However, controlling the size of $|x_{\mathbf{t}} y_{\mathbf{t}}|$ is surprisingly delicate as it requires a fine-grained understanding of the bouncing phase. The insight that guides the proof is the observation that during the bouncing phase, the GD iterates lie close to a certain envelope (Figure 9).

As a by-product of our analysis, we obtain a rigorous version of the quasi-static principle from which can more accurately track the sharpness gap and convergence rate (see Subsection C.5). The results of Theorem 1, Theorem 2, and Theorem 5 are displayed pictorially as Figure 9.

## 3 Understanding the bias evolution of the ReLU network

In this section, we use the insights from Section 2 to answer our main question, namely understanding the role of a large step size in learning threshold neurons for the ReLU network (2). Based on the observed dynamics (Figure 2), we can make our question more concrete as follows.

**Q.** What is the role of a large step size during the "*initial phase*" of training in which *(i)* the bias $b$ rapidly decreases and *(ii)* the sum of weights $a^- + a^+$ oscillates?

### 3.1 Approximating the initial phase of GD with the "mean model"

Deferring details to Appendix D, the GD dynamics for the ReLU network (2) in the initial phase are well-approximated by

$$\text{GD dynamics on } (a^-, a^+, b) \mapsto \ell_{\text{sym}}(d\,(a^- + a^+)\,g(b))\,,$$

where $\ell_{\text{sym}}(s) := \frac{1}{2}(\log(1 + \exp(-s)) + \log(1 + \exp(+s)))$ and $g(b) := \mathbb{E}_{z \sim \mathcal{N}(0,1)} \text{ReLU}(z + b)$ is the 'smoothed' ReLU. The GD dynamics can be compactly written in terms of the parameter $A_t := d\,(a_t^- + a_t^+)$.

$$A_{t+1} = A_t - 2d^2\eta\,\ell'_{\text{sym}}(A_t g(b_t))\,g(b_t)\,, \qquad b_{t+1} = b_t - \eta\,\ell'_{\text{sym}}(A_t g(b_t))\,A_t g'(b_t)\,. \quad (5)$$

We call these dynamics ***the mean model***. Figure 8 shows that the mean model closely captures the GD dynamics for the ReLU network (2), and we henceforth focus on analyzing the mean model.

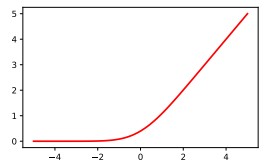

Figure 7: The 'smoothed' ReLU $g(b)$

The main advantage of the representation (5) is that it makes apparent the connection to the single-neuron example that we studied in Section 2. More specifically, (5) can be interpreted as the "rescaled" GD dynamics on the objective $(A, b) \mapsto \ell_{\mathrm{sym}}(Ag(b))$, where the step size for the $A$-dynamics is multiplied by $2d^2$. Due to this resemblance, we can apply the techniques from Section 2.

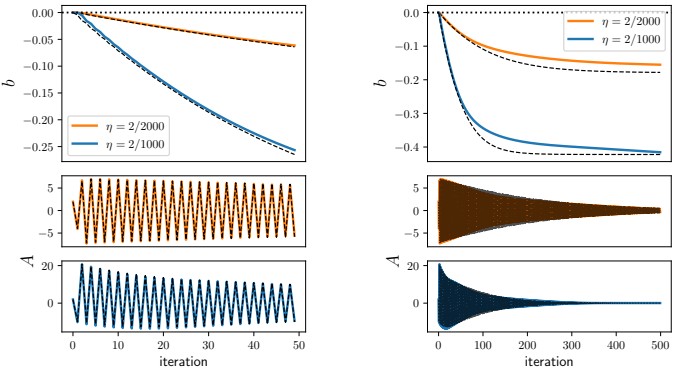

Figure 8: Under the same setting as Figure 1, we compare the mean model with the GD dynamics of the ReLU network. The mean model is plotted with black dashed line. Note that ***the mean model tracks the GD dynamics quite well during the initial phase of training.***

## 3.2 Two different regimes for the mean model

Throughout the section, we use the shorthand $\ell := \ell_{\mathrm{sym}}$, and focus on initializing wiht $a_0^\pm = \Theta(1/d)$, $a^- + a^+ \neq 0$, and $b_0 = 0$. This implies $A_0 = \Theta(1)$. We also note the following fact for later use.

**Lemma 2** (formula for the smoothed ReLU; see Subsection E.1). *The smoothed ReLU function $g$ can be expressed in terms of the PDF $\varphi$ and the CDF $\Phi$ of the standard Gaussian distribution as $g(b) = \varphi(b) + b\,\Phi(b)$. In particular, $g' = \Phi$.*

Note also that $b_t$ is monotonically decreasing. This is because $\ell'(A_t g(b_t))\, A_t g'(b_t) \geq 0$ since $\ell'$ is an odd function and $g(b), g'(b) > 0$ for any $b \in \mathbb{R}$.

Following Subsection 2.2, we begin with the continuous-time dynamics of the mean model:

$$\dot{A} = -2d^2\, \ell'(Ag(b))\, g(b), \qquad \dot{b} = -\ell'(Ag(b))\, Ag'(b). \tag{6}$$

**Lemma 3** (conserved quantity; see Subsection E.1). *Let $\kappa : \mathbb{R} \to \mathbb{R}$ be defined as $\kappa(b) := \int_0^b g/g'$. Along the gradient flow (6), the quantity $\frac{1}{2}A^2 - 2d^2\kappa(b)$ is conserved.*

Based on Lemma 3, if we initialize the continuous-time dynamics (6) at $(A_0, 0)$ and if $A_t \to 0$, then the limiting value of the bias $b_\infty^{\mathrm{GF}}$ satisfies $\kappa(b_\infty^{\mathrm{GF}}) = -\frac{1}{4d^2} A_0^2$, which implies that $b_\infty^{\mathrm{GF}} = -\Theta(\frac{1}{d^2})$; indeed, this holds since $\kappa'(0) = g(0)/g'(0) > 0$, so there exist constants $c_0, c_1 > 0$ such that $c_0 b \leq \kappa(b) \leq c_1 b$ for all $-1 \leq b \leq 0$. Since the mean model (5) tracks the continuous-time dynamics (6) until it reaches the $b$-axis, the mean model initialized at $(A_0, 0)$ also approximately reaches $(A_{t_0}, b_{t_0}) \approx (0, -\Theta(\frac{1}{d^2})) \approx (0, 0)$ in high dimension $d \gg 1$. In other words, ***the continuous-time dynamics*** (6) ***fails to learn threshold neurons.***

Once the mean model reaches the $b$-axis, we again identify two different regimes depending on the step size. A Taylor expansion of $\ell'$ around the origin yields the following approximate dynamics (here $\ell''(0) = 1/4$): $A_{t_0+1} \approx A_{t_0} - \frac{\eta d^2}{2} A_{t_0}\, g(b_{t_0})^2 = A_{t_0} \left(1 - \frac{\eta d^2}{2} g(b_{t_0})^2\right)$. We conclude that the condition which now dictates whether we have bouncing or convergence is $\frac{1}{2} d^2 g(b_{t_0})^2 > 2/\eta$.

*(i)* ***Gradient flow regime:*** If $2/\eta > d^2 g(0)^2/2 = d^2/(4\pi)$ (since $g(0)^2 = 1/(2\pi)$), i.e., the step size $\eta$ is *below* the threshold $8\pi/d^2$, then the final bias of the mean model $b_\infty^{\mathrm{MM}}$ satisfies $b_\infty^{\mathrm{MM}} \approx b_\infty^{\mathrm{GF}} \approx 0$. In other words, ***the mean model fails to learn threshold neurons***.

*(ii)* ***EoS regime:*** If $2/\eta < d^2/(4\pi)$, i.e., the step size $\eta$ is *above* the threshold $8\pi/d^2$, then $\frac{1}{2} d^2 g^2(b_\infty^{\mathrm{MM}}) < 2/\eta$, i.e., $b_\infty^{\mathrm{MM}} < g^{-1}(2/\sqrt{\eta d^2})$. For instance, if $\eta = \frac{10\pi}{d^2}$, then $b_\infty^{\mathrm{MM}} < -0.087$. In other words, ***the mean model successfully learns threshold neurons***.

### 3.3 Results for the mean model

**Theorem 3** (mean model, gradient flow regime; see Appendix E)**.** *Consider the mean model (5) initialized at $(A_0, 0)$, with step size $\eta = \frac{(8-\delta)\pi}{d^2}$ for some $\delta > 0$. Let $\gamma \coloneqq \frac{1}{200} \min\{\delta, 8-\delta, \frac{8-\delta}{|A_0|}\}$. Then, as long as $\eta \leq \gamma/|A_0|$, the limiting bias $b_\infty^{\mathrm{MM}}$ satisfies*

$$0 \geq b_\infty^{\mathrm{MM}} \geq -(\eta/\gamma)\,|A_0| = -O_{A_0,\delta}(1/d^2)\,.$$

*In other words, the mean model fails to learn threshold neurons.*

**Theorem 4** (mean model, EoS regime; see Appendix E)**.** *Consider the mean model initialized at $(A_0, 0)$, with step size $\eta = \frac{(8+\delta)\pi}{d^2}$ for some $\delta > 0$. Furthermore, assume that for all $t \geq 1$ such that $\frac{1}{2} d^2 g(b_t)^2 > 2/\eta$, we have $A_t \neq 0$. Then, the limiting bias $b_\infty^{\mathrm{MM}}$ satisfies*

$$b_\infty^{\mathrm{MM}} \leq g^{-1}\big(2/\sqrt{(8+\delta)\pi}\big) \leq -\Omega_\delta(1)\,.$$

*For instance, if $\eta = \frac{10\pi}{d^2}$, then $b_\infty^{\mathrm{MM}} < -0.087$. In other words, the mean model successfully learns threshold neurons.*

## 4 Conclusion

In this paper, we present the first explanation for the emergence of threshold neuron (i.e., ReLU neurons with negative bias) in models such as the sparse coding model (1) through a novel connection with the "edge of stability" (EoS) phenomenon. Along the way, we obtain a detailed and rigorous understanding of the dynamics of GD in the EoS regime for a simple class of loss functions, thereby shedding light on the impact of large learning rates in non-convex optimization.

Our approach is largely inspired by the recent paradigm of "*physics-style*" approaches to understanding deep learning based on simplified models and controlled experiments (c.f. (Zhang et al., 2022; von Oswald et al., 2023; Abernethy et al., 2023; Allen-Zhu and Li, 2023; Li et al., 2023; Ahn et al., 2023a,b)). We found such physics-style approach quite effective to understand deep learning, especially given the complexity of modern deep neural networks. We hope that our work inspires further research on understanding the working mechanisms of deep learning.

Many interesting questions remain, and we conclude with some directions for future research.

- ***Extending the analysis of EoS to richer models.*** Although the analysis we present in this work is restricted to simple models, the underlying principles can potentially be applied to more general settings. In this direction, it would be interesting to study models which capture the impact of the depth of the neural network on the EoS phenomenon. Notably, a follow-up work by Song and Yun (2023) uses bifurcation theory to extend our results to more complex models.

- ***The interplay between the EoS and the choice of optimization algorithm.*** As discussed in Subsection 2.3, the bouncing phase of the EoS substantially slows down the convergence of GD (see Figure 11). Investigating how different optimization algorithm (e.g., SGD, or GD with momentum) interact with the EoS phenomenon could potentially lead to practical speed-ups or improved generalization. Notably, a follow up work by Dai et al. (2023) studies the working mechanisms of a popular modern optimization technique called *sharpness-aware minimization* (Foret et al., 2021) based on our sparse coding problem.

- ***An end-to-end analysis of the sparse coding model.*** Finally, we have left open the motivating question of analyzing how two-layer ReLU networks learn to solve the sparse coding model (1). Despite the apparent simplicity of the problem, its analysis has thus far remained out of reach, and we believe that a resolution to this question would constitute compelling and substantial progress towards understanding neural network learning. We are hopeful that the insights in this paper provide the first step towards this goal.

## Acknowledgments

We thank Ronan Eldan, Suriya Gunasekar, Yuanzhi Li, Jonathan Niles-Weed, and Adil Salim for initial discussions on this project. KA was supported by the ONR grant (N00014-20-1-2394) and MIT-IBM Watson as well as a Vannevar Bush fellowship from Office of the Secretary of Defense. SC was supported by the NSF TRIPODS program (award DMS-2022448).

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

# Appendix

## A  Additional illustrations for Section 2

In this section, we provide some illustrations of the results presented in Section 2. We first illustrate the two different regimes of GD presented in Subsection 2.2.

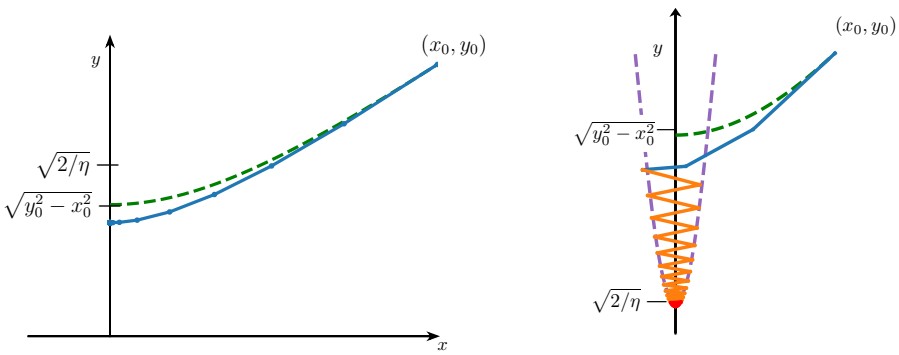

Figure 9: *Two regimes for GD.* We run GD on the square root loss with step size $\frac{1}{4}$. The gradient flow regime is illustrated on the left for $(x_0, y_0) = (3, 4)$. GD (blue) tracks the gradient flow (green) when $\eta < 2/(y_0^2 - x_0^2)$. Otherwise, as illustrated on the right for $(x_0, y_0) = (3, 6)$, GD is in the EoS regime and goes through a gradient flow phase (blue), an intermediate bouncing phase (orange) that tracks the quasi-static envelope (purple), and a converging phase (red).

Next, we next present detailed illustrations of the edge-of-stability regime depending on the choice of step size. Compare this plot with our theoretical results characterized in Theorem 2.

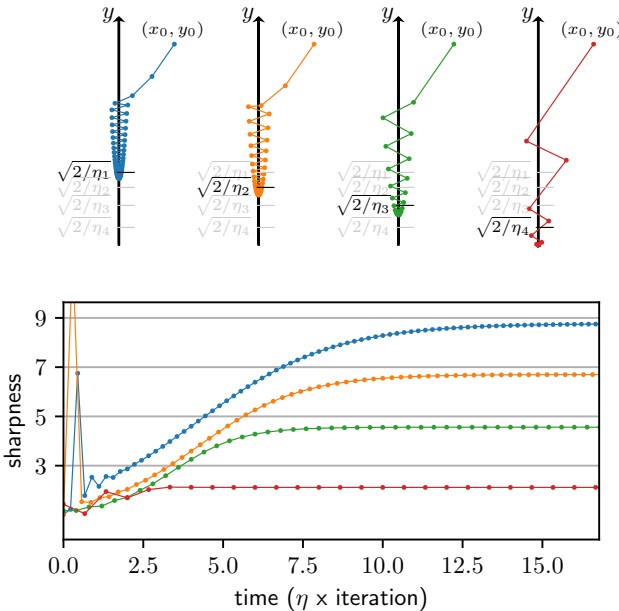

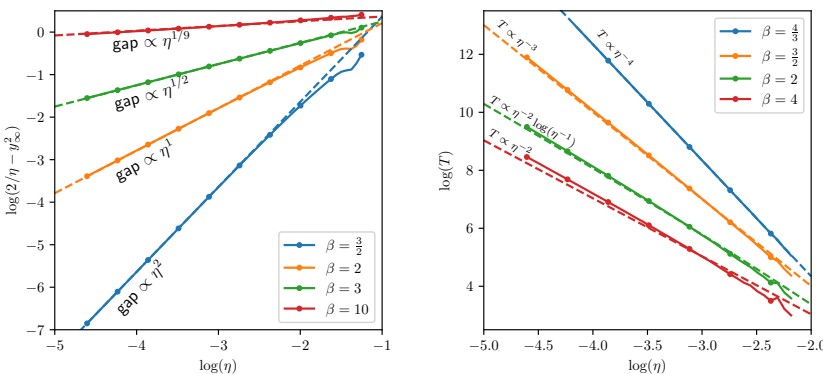

Figure 10: We plot the GD trajectory for $\ell(s) = \sqrt{1 + s^2}$ and sharpness for step sizes $2/\eta_1 = 9$, $2/\eta_2 = 7$, $2/\eta_3 = 5$, and $2/\eta_4 = 3$. In the EoS regime, the final sharpness is close to $2/(\text{step size})$.

Lastly, we present another detailed illustrations of Theorem 2 in terms of its dependence on $\beta$.

Figure 11: (Left) Log-log plot of the ***sharpness gap*** as a function of $\eta$, for $\ell_\beta$ in Example 1 and $\beta = \frac{3}{2}$, 2, 3, 10. (Right) Log-log plot of the ***iteration count*** for the bouncing region with $y_t^2 \in [\frac{2}{\eta}, \frac{3}{\eta}]$ as a function of $\eta$, for $\ell_\beta$ in Example 1 and $\beta = \frac{4}{3}$, $\frac{3}{2}$, 2, 4. ***The dashed lines show the predicted sharpness gap and iteration count*** with an offset computed via linear regression of the data for $\eta < e^{-2}$.

## B  Further experimental results

In this section, we report further experimental results which demonstrate that our theory, while limited to the specific models we study (namely, the single-neuron example and the mean model), is in fact

indicative of behaviors commonly observed in more realistic instances of neural network training. In particular, we show that threshold neurons often emerge in the presence of oscillations in the other weight parameters of the network.

## B.1    Experiments for the full sparse coding model

We provide the details for the top plot of Figure 3. consider the sparse coding model in the form (1). Compared to (2), we assume that the basis vectors are unknown, and the neural network learn them through additional parameters $\boldsymbol{W} = (\boldsymbol{w}_i)_{i=1}^{m}$ together with $m$ different weights $\boldsymbol{a} = (a_i)_{i=1}^{m}$ for the second layer as follows:

$$f(\boldsymbol{x}; \boldsymbol{a}, \boldsymbol{W}, b) = \sum_{i=1}^{m} a_i \operatorname{ReLU}\big(\langle \boldsymbol{w}_i, \boldsymbol{x} \rangle + b\big). \tag{7}$$

We show results for $d = 100$, $m = 2000$. We generate $n = 20000$ data points according to the aforementioned sparse coding model with $\lambda = 5$. We use the He initialization, i.e., $\boldsymbol{a} \sim \mathcal{N}(0, I_m/m)$, $\boldsymbol{w} \sim \mathcal{N}(0, I_d/d)$, and $b = 0$. As shown in the top plot of Figure 3, the bias decreases more with the large learning rate. Further, we report the behavior of the average of second layer weights in Figure 12 (left), and confirm that the sum oscillates.

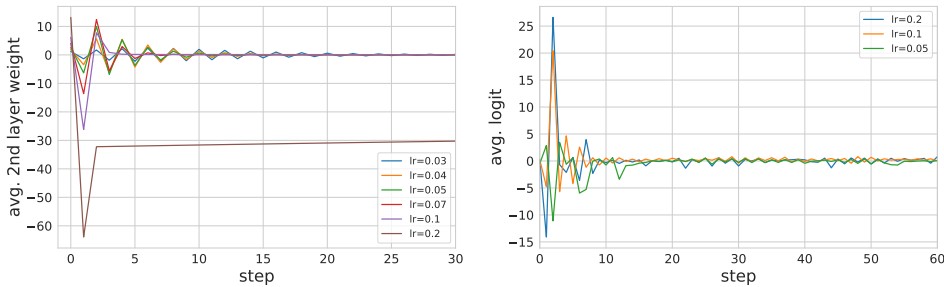

Figure 12: (*Left*) The average of the second layer weights of the ReLU network (7). Note that the average value oscillates similarly to our findings for the mean model. (*Right*) *Oscillation of logit of ResNet18 model averaged over the (binary) CIFAR-10 training set.* Since the dataset is binary, the logit is simply a scalar.

## B.2    Experiments on the CIFAR-10 dataset

Next, we provide the details for the bottom plot of Figure 3. We train ResNet-18 on a binarized version of the CIFAR-10 dataset formed by taking only the first two classes; this is done for the purpose of monitoring the average logit of the network. The average logit is measured over the entire training set. The median bias is measured at the last convolutional layer right before the pooling. For the optimizer, we use full-batch GD with no momentum or weight decay, plus a cosine learning rate scheduler where learning rates shown in the plots are the initial values.

***Oscillation of expected output (logit) of the network.*** Bearing a striking resemblance to our two-layer models, as one can see from Figure 12 (right) that the expected mean of the output (logit) of the deep net also oscillates due to GD dynamics. As we have argued in the previous sections, this occurs as the bias parameters are driven towards negative values.

***Results for SGD.*** In Figure 13, we report qualitatively similar phenomena when we instead train ResNet-18 with stochastic gradient descent (SGD), where we use all ten classes of CIFAR-10. Again, the median bias is measured at the last convolutional layer. We further report the average activation which is the output of the ReLU activation at the last convolutional layer, averaged over the neurons and the entire training set. The average activation statistics represent the hidden representations before the linear classifier part, and lower values represent sparser representations. Interestingly, the threshold neuron also emerges with larger step sizes similarly to the case of gradient descent.

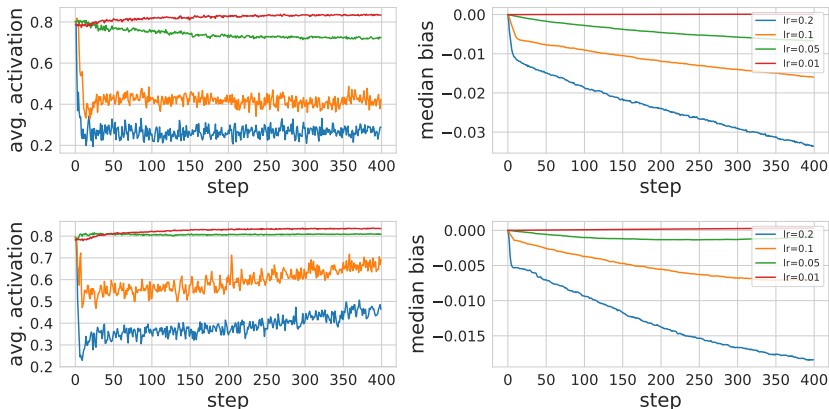

Figure 13: SGD dynamics of ResNet-18 on (multiclass) CIFAR-10 with various learning rates and batch sizes. (Top) batch size 100; (Bottom) batch size 1000. The results are consistent across different batch sizes.

# C   Proofs for the single-neuron linear network

We start by describing basic and relevant properties of the model and the assumptions on $\ell$.

**Basic properties.** If $\ell$ is minimized at $0$, then the *global minimizers* of $f$ are the $x$- and $y$-axes. The *gradient and Hessian* of $f$ are given by:

$$\nabla f(x, y) = \ell'(xy) \begin{bmatrix} y \\ x \end{bmatrix},$$

$$\nabla^2 f(x, y) = \ell''(xy) \begin{bmatrix} y \\ x \end{bmatrix}^{\otimes 2} + \ell'(xy) \begin{bmatrix} 0 & 1 \\ 1 & 0 \end{bmatrix}. \tag{8}$$

This results in GD *iterates* $x_t, y_t$, for step size $\eta > 0$ and iteration $t \geq 0$:

$$x_{t+1} = x_t - \eta\, \ell'(x_t y_t)\, y_t,$$
$$y_{t+1} = y_t - \eta\, \ell'(x_t y_t)\, x_t.$$

**Lemma 4** (invariant lines). *Assume that $\ell$ is even, so that $\ell'$ is odd. Then, the lines $y = \pm x$ are invariant for gradient descent on $f$.*

*Proof.* If $y_t = \pm x_t$, then

$$y_{t+1} = y_t - \eta\, \ell'(x_t y_t)\, x_t = \pm x_t \mp \eta\, \ell'(x_t^2)\, x_t,$$
$$x_{t+1} = x_t - \eta\, \ell'(x_t y_t)\, y_t = x_t - \eta\, \ell'(x_t^2)\, x_t,$$

and hence $y_{t+1} = \pm x_{t+1}$. Note that the iterates $(x_t)_{t\geq 0}$ are the iterates of GD with step size $\eta$ on the one-dimensional loss function $x \mapsto \frac{1}{2}\, \ell(x^2)$. $\qquad\square$

We focus instead on initializing away from these two lines. We now state our assumptions on $\ell$.

We gather together some elementary properties of $\ell$.

**Lemma 5** (properties of $\ell$). *Suppose that Assumption (A1) holds.*

1. *$\ell$ is minimized at the origin and $\ell'(0) = 0$.*
2. *Suppose that $\ell$ is four times continuously differentiable near the origin. If Assumption (A2) holds, then $\ell^{(4)}(0) \leq 0$. Conversely, if $\ell^{(4)}(0) < 0$, then Assumption (A2) holds for $\beta = 2$.*

*Proof.* The first statement is straightforward. The second statement follows from Taylor expansion: for $s \neq 0$ near the origin,

$$\frac{\ell'(s)}{s} = \frac{\ell'(0) + \ell''(0)\, s + \int_0^s (s - r)\, \ell'''(r)\, \mathrm{d}r}{s} = 1 + \int_0^s \left(1 - \frac{r}{s}\right) \ell'''(r)\, \mathrm{d}r. \tag{9}$$

Since $\ell'''$ is odd, then Assumption (A2) and (9) imply that $\ell'''$ is non-positive on $(0, \varepsilon)$ for some $\varepsilon > 0$, which in turn implies $\ell^{(4)}(0) \leq 0$. Conversely, if $\ell^{(4)}(0) < 0$, then there exists $\varepsilon > 0$ such that $\ell'''(s) \leq -\varepsilon s$ for $s \in (0, \varepsilon)$. From (9), we see that $\ell'(s)/s \leq 1 - \varepsilon \int_0^s (s - r)\, dr \leq 1 - \varepsilon s^2/2$. By symmetry, we conclude that Assumption (A2) holds with $\beta = 2$ and some $c > 0$. $\qquad\square$

We give some simple examples of losses satisfying our assumptions.

**Example 1.** The examples below showcase several functions $\ell$ that satisfy Assumptions (A1) and (A2) with different values of $\beta$.

- *Rescaled and symmetrized logistic loss.* $\ell_{\mathrm{rsym}}(s) := \frac{1}{2}\, \ell_{\mathrm{logi}}(-2s) + \frac{1}{2}\, \ell_{\mathrm{logi}}(+2s)$.
  Note $\ell'_{\mathrm{rsym}}(s) = \tanh(s)$, thus $\ell'_{\mathrm{rsym}}(s)/s \leq 1$ and $\ell'_{\mathrm{rsym}}(s)/s \leq 1 - \frac{1}{4}|s|^2$, for $|s| < \frac{1}{4}$.
- *Square root loss.* $\ell_{\mathrm{sqrt}}(s) := \sqrt{1 + s^2}$.
  Note $\ell'_{\mathrm{sqrt}}(s) = \frac{s}{\sqrt{1+s^2}}$, thus $\ell'_{\mathrm{sqrt}}(s)/s \leq 1$ and $\ell'_{\mathrm{sqrt}}(s)/s \leq 1 - \frac{2}{5}|s|^2$, for $|s| < \frac{2}{5}$.
- *Huber loss.* $\ell_{\mathrm{Hub}}(s) := \frac{s^2}{2}\, \mathbb{1}\{s \in [-1, 1]\} + \left(|s| - \frac{1}{2}\right) \mathbb{1}\{s \notin [-1, 1]\}$.
  Note $\ell'_{\mathrm{Hub}}(s) = s\, \mathbb{1}\{s \in [-1, 1]\} + \mathrm{sgn}(s)\, \mathbb{1}\{s \notin [-1, 1]\}$, thus $\ell'_{\mathrm{Hub}}(s)/s \leq 1$, i.e., we have Assumption (A2) with $\beta = +\infty$.
- *Higher-order.* For $\beta > 1$ let $c_\beta := \frac{1}{\beta+1}\left(\frac{\beta}{\beta+1}\right)^\beta$ and $r_\beta := \frac{\beta+1}{\beta}$. We define $\ell_\beta$ implicitly via its derivative
$$\ell'_\beta(s) := s\left(1 - c_\beta\,|s|^\beta\right) \mathbb{1}\{s^2 < r_\beta^2\} + \mathrm{sgn}(s)\, \mathbb{1}\{s^2 \geq r_\beta^2\}.$$

  By definition, $\ell'_\beta(s)/s \leq 1$ and $\ell'_\beta(s)/s \leq 1 - c_\ell\,|s|^\beta$, where $c_\ell = c_\beta \wedge r_\beta$.

We now prove our main results from Subsection 2.3 in order.

## C.1 Approximate conservation along GD

We begin by stating and proving the approximate conservation of $y^2 - x^2$ for the GD dynamics.

**Lemma 6** (approximately conserved quantity). *Let $(\tilde{x}, \tilde{y}) \in \mathbb{R}^2$ be such that $\tilde{y} > \tilde{x} > 0$ with $\tilde{y}^2 - \tilde{x}^2 = 1$. Suppose that we run GD on $f$ with step size $\eta$ with initial point $(x_0, y_0) := \sqrt{\frac{\gamma}{\eta}}\,(\tilde{x}, \tilde{y})$, for some $\gamma > 0$. Then, there exists $t_0 = O(\frac{1}{\eta})$ such that $\sup_{t \geq t_0} |x_t| \leq O(\sqrt{(\gamma^{-1} \vee \gamma)\,\eta})$ and*
$$y_{t_0}^2 - x_{t_0}^2 = \left(1 - O(\eta)\right)\left(y_0^2 - x_0^2\right),$$
*where the implied constant depends on $\tilde{x}$, $\tilde{y}$, and $\ell$.*

*Proof.* Let $D_t := y_t^2 - x_t^2$ and note that
$$\begin{aligned}
D_{t+1} &= \left(y_t - \eta\, \ell'(x_t y_t)\, x_t\right)^2 - \left(x_t - \eta\, \ell'(x_t y_t)\, y_t\right)^2 \\
&= \left(1 - \eta^2\, \ell'(x_t y_t)^2\right) D_t\,.
\end{aligned}$$
Since $\ell$ is 1-Lipschitz, then $D_{t+1} = (1 - O(\eta^2))\, D_t$.

This shows that for $t \lesssim 1/\eta^2$, we have $y_t^2 - x_t^2 = D_t \gtrsim D_0 = y_0^2 - x_0^2 \asymp \gamma/\eta$. Since $\ell''(0) = 1$, there exist constants $c_0, c_1 > 0$ such that $\ell'(|xy|) \geq \ell'(c_0) \geq c_1$ whenever $|xy| \geq c_0$. Hence, for all $t \geq 1$ such that $t \lesssim 1/\eta^2$, $x_t > 0$, and $|x_t y_t| \geq c_0$, we have $y_t^2 \gtrsim \gamma/\eta$ and
$$x_{t+1} = x_t - \eta\, \ell'(x_t y_t)\, y_t = x_t - \Theta(\eta y_t) = x_t - \Theta(\sqrt{\gamma\eta})\,. \tag{10}$$

Since $x_0 \asymp \sqrt{\gamma/\eta}$, this shows that after at most $O(1/\eta)$ iterations, we must have either $x_t < 0$ or $|x_t y_t| \leq c_0$ for the first time. In the first case, (10) shows that $|x_t| \lesssim \sqrt{\gamma\eta}$. In the second case, since $y_t^2 \gtrsim \gamma/\eta$, we have $|x_t| \lesssim \sqrt{\eta/\gamma}$. Let $t_0$ denote the iteration at which this occurs.

Next, for iterations $t \geq t_0$, we use the dynamics (10) for $x$ and the fact that $\ell'(x_t y_t)$ has the same sign as $x_t$ to conclude that there are two possibilities: either $x_{t+1}$ has the same sign as $x_t$, in which case $|x_{t+1}| \leq |x_t|$, or $x_{t+1}$ has the opposite sign as $x_t$, in which case $|x_{t+1}| \leq \eta\, |\ell'(x_t y_t)|\, y_t \leq \eta y_t \leq O(\sqrt{\gamma\eta})$. This implies $\sup_{t \geq t_0} |x_t| \leq O(\sqrt{(\gamma^{-1} \vee \gamma)\,\eta})$ as asserted. $\qquad\square$

## C.2 Gradient flow regime

In this section, we prove Theorem 1. From Lemma 6, there exists an iteration $t_0$ such that $|x_{t_0}| \lesssim \sqrt{\eta/(2-\delta)}$ and

$$\frac{2-\delta}{\eta} - O(2-\delta) \leq y_{t_0}^2 \leq \frac{2-\delta}{\eta} + x_{t_0}^2 \leq \frac{2-\delta}{\eta} + O\left(\frac{\eta}{2-\delta}\right).$$

In particular, $C := |x_{t_0} y_{t_0}| \lesssim 1$.

We prove by induction the following facts: for $t \geq t_0$,

1. $|x_t y_t| \leq C$.
2. $|x_t| \leq |x_{t_0}| \exp(-\Omega(\alpha\,(t-t_0)))$, where $\alpha := \min\{\delta, 2-\delta\}$.

Suppose that these conditions hold up to iteration $t \geq t_0$. By Assumption (A2), we have $|\ell'(s)| \leq |s|$ for all $s \neq 0$. Therefore,

$$y_{t+1} = y_t - \eta\,\ell'(x_t y_t)\,x_t \geq (1 - \eta x_t^2)\,y_t$$

$$\geq \exp\left(-O\left(\frac{\eta^2}{2-\delta}\right) \exp\left(-\Omega(\alpha\,(t-t_0))\right)\right) y_t$$

$$\geq \exp\left(-O\left(\frac{\eta^2}{2-\delta}\right) \sum_{s=t_0}^{t} \exp\left(-\Omega(\alpha\,(s-t_0))\right)\right) y_{t_0} \geq \exp\left(-O\left(\frac{\eta^2}{\alpha\,(2-\delta)}\right)\right) y_{t_0},$$

$$y_{t+1}^2 \geq \frac{2-\delta}{\eta} - O(2-\delta) - O\left(\frac{\eta}{\alpha}\right). \tag{11}$$

In particular, $\frac{1}{2}\frac{2-\delta}{\eta} \leq y_t^2 \leq \frac{2-\delta/2}{\eta}$ throughout. In order for these assertions to hold, we require $\eta^2 \lesssim \alpha\,(2-\delta)$, i.e., $\eta \lesssim \min\{\sqrt{\delta}, 2-\delta\}$.

Next, we would like to show that $t \mapsto |x_t|$ is decaying exponentially fast. Since

$$|x_{t+1}| = |x_t - \eta\,\ell'(x_t y_t)\,y_t| = \big|\,|x_t| - \eta\,\ell'(|x_t|\,y_t)\,y_t\,\big|,$$

it suffices to consider the case when $x_t > 0$. Assumption (A2) implies that

$$x_{t+1} \geq (1 - \eta y_t^2)\,x_t \geq -\left(1 - \frac{\delta}{2}\right) x_t.$$

For the upper bound, we split into two cases. We begin by observing that since $\ell$ is twice continuously differentiable near the origin with $\ell''(0) = 1$, there is a constant $\varepsilon_0$ such that $|s| < \varepsilon_0$ implies $|\ell'(s)| \geq \frac{1}{2}|s|$. If $s_t := x_t y_t \leq \varepsilon_0$, then

$$x_{t+1} \leq \left(1 - \frac{\eta}{2}\,y_t^2\right) x_t \leq \left(1 - \frac{2-\delta}{4}\right) x_t.$$

Otherwise, if $s_t \geq \varepsilon_0$, then

$$x_{t+1} \leq x_t - \eta\,\ell'(\varepsilon_0)\,y_t \leq x_t - \eta\,\ell'(\varepsilon_0)\,\frac{y_t^2}{s_t} \leq x_t - \eta\,\ell'(\varepsilon_0)\,\frac{2-\delta}{2C\eta}\,x_t \leq \left(1 - \Omega(2-\delta)\right) x_t.$$

Combining these inequalities, we obtain

$$|x_{t+1}| \leq |x_t| \exp\left(-\Omega(\alpha)\right).$$

This verifies the second statement in the induction. The first statement follows because both $t \mapsto |x_t|$ and $t \mapsto y_t$ are decreasing.

This shows in particular that $|x_t| \searrow 0$, i.e., we have global convergence. To conclude the proof, observe that (11) gives a bound on the final sharpness.

*Remark* 1. The proof also gives us estimates on the convergence rate. Namely, from Lemma 6, the initial phase in which we approach the $y$-axis takes $O(\frac{1}{\eta})$ iterations. For the convergence phase, in order to achieve $\varepsilon$ error, we need $|x_t| \lesssim \frac{\sqrt{\varepsilon\eta}}{\sqrt{2-\delta}}$; hence, the convergence phase needs only $O(\frac{1}{\alpha}\log\frac{1}{\varepsilon})$ iterations. Note that the rate of convergence in the latter phase does not depend on the step size $\eta$.

## C.3 EoS regime: proof outline

We give a brief outline of the proof of Theorem 2: As before, Lemma 6 shows that GD reaches the $y$-axis approximately at $(0, \sqrt{y_0^2 - x_0^2})$. At this point, $x$ starts bouncing while $y$ steadily decreases, and we argue that unless $x_t = 0$ or $y_t^2 \leq 2/\eta$, the GD dynamics cannot stabilize (see Lemma 7).

To bound the gap $2/\eta - y_\infty^2$, we look at the first iteration $\mathbf{t}$ such that $y_{\mathbf{t}}^2$ crosses $2/\eta$. By making use of Assumption (A2), we simultaneously control both the convergence rate of $|x_t|$ to zero and the decrease in $y_t^2$ in order to prove that

$$y_\infty^2 \geq \frac{2}{\eta} - O(|x_{\mathbf{t}} y_{\mathbf{t}}|), \tag{12}$$

see Proposition 1. Therefore, to establish Theorem 2, we must bound $|x_{\mathbf{t}} y_{\mathbf{t}}|$ at iteration $\mathbf{t}$.

Controlling the size of $|x_{\mathbf{t}} y_{\mathbf{t}}|$, however, is surprisingly delicate as it requires a fine-grained understanding of the bouncing phase. The insight that guides the proof is the observation that during the bouncing phase, the GD iterates lie close to a certain envelope (Figure 9). This envelope is predicted by the quasi-static heuristic as described in Ma et al. (2022). Namely, suppose that after one iteration of GD, we have perfect bouncing: $x_{t+1} = -x_t$. Substituting this into the GD dynamics, we obtain the equation

$$\eta \ell'(x_t y_t) y_t = 2 x_t. \tag{13}$$

According to Assumption (A2), we have $\ell'(x_t y_t) = x_t y_t (1 - \Omega(|x_t y_t|^\beta))$, Together with (13), if $y_t^2 = (2 + \delta_t)/\eta \geq 2/\eta$, where $\delta_t$ is sufficiently small, it suggests that

$$|x_t y_t| \lesssim \delta_t^{1/\beta}. \tag{14}$$

The quasi-static prediction (14) fails when $\delta_t$ is too small. Nevertheless, we show that it remains accurate as long as $\delta_t \gtrsim \eta^{\beta/(\beta-1)}$, and consequently we obtain $|x_{\mathbf{t}} y_{\mathbf{t}}| \lesssim \eta^{1/(\beta-1)}$. Combined with (12), it yields Theorem 2.

## C.4 EoS regime: crossing the threshold and the convergence phase

In this section, we prove Theorem 2. We first show that $y_t^2$ must cross $2/\eta$ in order for GD to converge, and we bound the size of the jump across $2/\eta$ once this happens.

Throughout this section and the next, we use the following notation:

- $s_t := x_t y_t$;
- $r_t := \ell'(s_t)/s_t$.

In this notation, we can write the GD equations as

$$x_{t+1} = (1 - \eta r_t y_t^2) x_t,$$
$$y_{t+1} = (1 - \eta r_t x_t^2) y_t.$$

We also make a remark regarding Assumption (A2). If $\beta < +\infty$, then Assumption (A2) is equivalent to the following seemingly strongly assumption: for all $r > 0$, there exists a constant $c(r) > 0$ such that

$$\frac{\ell'(s)}{s} \leq 1 - c(r) |s|^\beta, \qquad \text{for all } 0 < |s| \leq r. \tag{A2$^+$}$$

Indeed, Assumption (A2) states that (A2$^+$) holds for *some* $r > 0$. To verify that (A2$^+$) holds for some larger $r' > r$, we can split into two cases. If $|s| \leq r$, then $\ell'(s)/s \leq 1 - c |s|^\beta$. Otherwise, if $|s| > r$, then $\ell'(r)/r < 1$ and the 1-Lipschitzness of $\ell'$ imply that $\ell'(s)/s < 1$ for $r \leq |s| \leq r'$, and hence $\ell'(s)/s \leq 1 - c' |s|^\beta$, for a sufficiently small constant $c' > 0$; thus we can take $c(r') = c \wedge c'$. Later, we will invoke (A2$^+$) with $r$ chosen to be a universal constant, so that $c(r)$ can also be thought of as universal.

We begin with the following result about the limiting value of $y_t$.

**Lemma 7** (threshold crossing). *Let $(\tilde{x}, \tilde{y}) \in \mathbb{R}^2$ satisfy $\tilde{y} > \tilde{x} > 0$ with $\tilde{y}^2 - \tilde{x}^2 = 1$. Suppose we initialize GD with step size $\eta$ with initial point $(x_0, y_0) := \sqrt{\frac{2+\delta}{\eta}} (\tilde{x}, \tilde{y})$, where $\delta > 0$ is a constant. Then either $x_t = 0$ for some $t$ or*

$$\lim_{t \to \infty} y_t^2 \leq \frac{2}{\eta} .$$

*Proof.* Assume throughout that $x_t \neq 0$ for all $t$. Recall the dynamics for $y$:

$$y_{t+1} = y_t - \eta \, \ell'(x_t y_t) \, x_t .$$

By assumption $\ell'(s)/s \to 1$ as $s \to 0$, and $\ell'$ is increasing, so this equation implies that if $\liminf_{t \to \infty} |x_t| > 0$ then $y_t^2$ must eventually cross $2/\eta$.

Suppose for the sake of contradiction that there exists $\varepsilon > 0$ with $y_t^2 > (2 + \varepsilon)/\eta$, for all $t$. Let $\varepsilon' > 0$ be such that $1 - (2 + \varepsilon)(1 - \varepsilon') < -1$, i.e., $\varepsilon' < \frac{\varepsilon}{2+\varepsilon}$. Then, there exists $\delta > 0$ such $|x_t| \leq \delta$ implies $r_t > 1 - \varepsilon'$, hence

$$\frac{|x_{t+1}|}{|x_t|} = |1 - \eta r_t y_t^2| > |(2 + \varepsilon)(1 - \varepsilon') - 1| > 1 .$$

The above means that $|x_t|$ increases until it exceeds $\delta$, i.e., $\liminf_{t \to \infty} |x_t| \geq \delta$. This is our desired contradiction and it implies that $\lim_{t \to \infty} y_t^2 \leq 2/\eta$. $\qquad\square$

**Lemma 8** (initial gap). *Suppose that at some iteration $\mathbf{t}$, we have*

$$y_{\mathbf{t}+1}^2 < \frac{2}{\eta} \leq y_{\mathbf{t}}^2 .$$

*Then, it holds that*

$$y_{\mathbf{t}+1}^2 \geq \frac{2}{\eta} - 2\eta s_{\mathbf{t}}^2 .$$

*Proof.* We can bound

$$y_{\mathbf{t}+1}^2 = y_{\mathbf{t}}^2 - 2\eta \, \ell'(x_{\mathbf{t}} y_{\mathbf{t}}) \, x_{\mathbf{t}} y_{\mathbf{t}} + \eta^2 \, \ell'(x_{\mathbf{t}} y_{\mathbf{t}})^2 \, x_{\mathbf{t}}^2 \geq y_{\mathbf{t}}^2 - 2\eta \, |x_{\mathbf{t}} y_{\mathbf{t}}|^2 ,$$

where we used the fact that $|\ell'(s)| \leq |s|$ for all $s \in \mathbb{R}$, $\qquad\square$

The above lemma shows that the size of the jump across $2/\eta$ is controlled by the size of $|s_{\mathbf{t}}|$ at the time of the crossing. From Lemma 6, we know that $|s_{\mathbf{t}}| \lesssim 1$, where the implied constant depends on $\delta$. Hence, the size of the jump is always $O(\eta)$.

We now provide an analysis of the convergence phase, i.e., after $y_t^2$ crosses $2/\eta$.

**Proposition 1** (convergence phase). *Suppose that $y_{\mathbf{t}}^2 < 2/\eta \leq y_{\mathbf{t}-1}^2$. Then, GD converges to $(0, y_\infty)$ satisfying*

$$\frac{2}{\eta} - O(|s_{\mathbf{t}}|) \leq y_\infty^2 \leq \frac{2}{\eta} .$$

*Proof.* Write $y_t^2 = (2 - \rho_t)/\eta$, so that $\rho_t = 2 - \eta y_t^2$. We write down the update equations for $x$ and for $\rho$. First, by the same argument as in the proof of Theorem 1, we have

$$|x_{t+1}| \leq |x_t| \exp(-\Omega(\rho_t)) . \tag{15}$$

Next, using $r_t \leq 1$,

$$y_{t+1} = (1 - \eta r_t x_t^2) \, y_t \geq (1 - \eta x_t^2) \, y_t ,$$
$$y_{t+1}^2 \geq (1 - 2\eta x_t^2) \, y_t^2 ,$$

which translates into

$$\rho_{t+1} \leq \rho_t + 2\eta^2 x_t^2 y_t^2 \leq \rho_t + 4\eta x_t^2 . \tag{16}$$

Using these two inequalities, we can conclude as follows. Let $q > 0$ be a parameter chosen later, and let $t$ be the first iteration for which $\rho_t \geq q$ (if no such iteration exists, then $\rho_t \leq q$ for all $t$). Note that $\rho_t \leq q + O(\eta |x_\mathbf{t}|)$ due to (15) and (16). By (15), we conclude that for all $t' \geq t$,

$$|x_{t'}| \leq |x_t| \exp\big(-\Omega(q\,(t'-t))\big) \leq |x_\mathbf{t}| \exp\big(-\Omega(q\,(t'-t))\big).$$

Substituting this into (16),

$$\rho_{t'} \leq \rho_t + 4\eta \sum_{s=t}^{t'-1} x_s^2 \leq q + O(\eta |x_\mathbf{t}|) + O(\eta |x_\mathbf{t}|^2) \sum_{s=1}^{t'-1} \exp\big(-\Omega(q\,(s-t))\big)$$

$$\leq q + O(\eta |x_\mathbf{t}|) + O\big(\frac{\eta |x_\mathbf{t}|^2}{q}\big).$$

By optimizing this bound over $q$, we find that for all $t$,

$$\rho_t \lesssim \sqrt{\eta}\,|x_\mathbf{t}| \lesssim \eta\,|s_\mathbf{t}|.$$

Translating this result back into $y_t^2$ yields the result. $\qquad\square$

Let us take stock of what we have established thus far.

- According to Lemma 6, $|s_t|$ is bounded for all $t$ by a constant.
- Then, from Lemma 7 and Lemma 8, we must have either $y_t^2 \to 2/\eta$, or $2/\eta - O(\eta) \leq y_\mathbf{t}^2 \leq 2/\eta$ for some iteration $\mathbf{t}$.
- In the latter case, Proposition 1 shows that the limiting sharpness is $2/\eta - O(1)$.

Note also that the analyses thus far have not made use of Assumption (A2), i.e., we have established the $\beta = +\infty$ case of Theorem 2. Moreover, for all $\beta > 1$, the asymptotic $2/\eta - O(1)$ still shows that the limiting sharpness is close to $2/\eta$, albeit with suboptimal rate. The reader who is satisfied with this result can then skip ahead to subsequent sections. The remainder of this section and the next section are devoted to substantial refinements of the analysis.

To see where improvements are possible, note that both Lemma 8 and Proposition 1 rely on the size of $|s_\mathbf{t}|$ at the crossing. Our crude bound of $|s_\mathbf{t}| \lesssim 1$ does not capture the behavior observed in experiments, in which $|s_\mathbf{t}| \lesssim \eta^{1/(\beta-1)}$. By substituting this improved bound into Lemma 7, we would deduce that the gap at the crossing is $O(\eta^{1+2/(\beta-1)})$, and then Proposition 1 would imply that the limiting sharpness is $2/\eta - O(\eta^{1/(\beta-1)})$. Another weakness of our proof is that it provides nearly no information about the dynamics during the bouncing phase, which constitutes an incomplete understanding of the EoS phenomenon. In particular, we experimentally observe that during the bouncing phase, the iterates lie very close to the quasi-static envelope (Figure 9). In the next section, we will rigorously prove all of these observations.

Before doing so, however, we show that Proposition 1 can be refined by using Assumption (A2), which could be of interest in its own right. It shows that even if the convergence phase begins with a large value of $|s_\mathbf{t}|$, the limiting sharpness can be much closer to $2/\eta$ than what Proposition 1 suggests. The following proposition combined with Lemma 6 implies Theorem 2 for all $\beta > 2$, but it is insufficient for the case $1 < \beta \leq 2$. From now on, we assume $\beta < +\infty$.

**Proposition 2** (convergence phase; refined). *Suppose that $y_\mathbf{t}^2 < 2/\eta \leq y_{\mathbf{t}-1}^2$. Then, GD converges to $(0, y_\infty)$ satisfying*

$$\frac{2}{\eta} \geq y_\infty^2 \geq \frac{2}{\eta} - O(\eta |s_\mathbf{t}|^2) - \begin{cases} O(\eta^{1/(\beta-1)}), & \beta > 2, \\ O(\eta \log(|s_\mathbf{t}|/\eta)), & \beta = 2, \\ O(\eta |s_\mathbf{t}|^{2-\beta}), & \beta < 2. \end{cases}$$

*Proof.* Let $y_t^2 = (2 - \rho_t)/\eta$ as before. We quantify the decrease of $|x_t|$ in terms of $\rho_t$ and conversely the increase of $\rho_t$ in terms of $|x_t|$ by tracking the half-life of $|x_t|$, i.e., the number of iterations it takes $|x_t|$ to halve. We call these epochs: at the $i$-th epoch, we have

$$2^{-(i+1)}\sqrt{\eta} < |x_t| \leq 2^{-i}\sqrt{\eta}.$$

Let $i_0$ be the index of the first epoch, i.e., $i_0 = \lfloor \log_2(\sqrt{\eta}/|x_\mathbf{t}|) \rfloor$. Due to Lemma 6, we know that $i_0 \geq -O(1)$. From (15), $|x_t|$ is monotonically decreasing and consequently $|s_t|$ is decreasing as

well. Also, our bound on the limiting sharpness implies that $y_t^2 > 1/\eta$ for all $t$, provided that $\eta$ is sufficiently small.

Let us now compute the dynamics of $\rho_t$ and $|x_t|$. At epoch $i$, $|x_t| > 2^{-(i+1)}\sqrt{\eta}$ hence $|s_t| > 2^{-(i+1)}$. Assumption (A2$^+$) with $r = |s_\mathbf{t}| \lesssim 1$ implies that

$$\frac{\ell'(s_t)}{s_t} \leq 1 - c\, 2^{-\beta\,(i+1)}, \tag{17}$$

where $c = c(|s_\mathbf{t}|)$. This allows to refine (15) on the decrease of $|x_t|$ to

$$\frac{|x_{t+1}|}{|x_t|} = \eta r_t y_t^2 - 1 \leq (2 - \rho_t)\left(1 - c\,2^{-\beta\,(i+1)}\right) - 1 \leq 1 - \rho_t - c\,2^{-\beta\,(i+1)},$$

where the first inequality follows from (17) and the second from $\rho_t = 2 - \eta y_t^2 < 1$. In turn, this inequality shows that the $i$-th phase only requires $O(2^{\beta i})$ iterations.

Hence, if $t(i)$ denotes the start of the $i$-th epoch, then (16) shows that

$$\rho_{t(i+1)} \leq \rho_{t(i)} + 4\eta^2 \cdot 2^{-2i} \cdot O(2^{\beta i}) \leq \rho_{t(i)} + O(\eta^2\, 2^{(\beta-2)\,i}).$$

Summing this up, we have

$$\rho_{t(i)} \leq \rho_\mathbf{t} + \eta^2 \times \begin{cases} O(2^{(\beta-2)\,i}), & \beta > 2, \\ O(i - i_0), & \beta = 2, \\ O(2^{(\beta-2)\,i_0}) = O(|s_\mathbf{t}|^{2-\beta}), & \beta < 2. \end{cases}$$

In the case of $\beta < 2$, the final sharpness satisfies $2/\eta - O(\rho_\mathbf{t}/\eta) - O(\eta\,|s_\mathbf{t}|^{2-\beta}) \leq y_\infty^2 \leq 2/\eta$.

In the other two cases, suppose that we use this argument until epoch $i_\star$ such that $2^{-i_\star} \asymp \eta^\gamma$. Then, we have $|x_{t(i_\star)}| \asymp \eta^{\gamma+1/2}$, $|s_{t(i_\star)}| \asymp \eta^\gamma$, and by using the argument from Proposition 1 from iteration $t(i_\star)$ onward we obtain

$$\rho_\infty = \rho_{t(i_\star)} + \rho_\infty - \rho_{t(i_\star)} \leq \rho_\mathbf{t} + O(\eta^{\gamma+1}) + \eta^2 \times \begin{cases} O(2^{(\beta-2)\,i_\star}) = O(\eta^{-\gamma\,(\beta-2)}), & \beta > 2, \\ O(i_\star - i_0), & \beta = 2. \end{cases}$$

We optimize over the choice of $\gamma$, obtaining $\gamma = 1/(\beta - 1)$ and thus

$$\rho_\infty \leq \rho_\mathbf{t} + \begin{cases} O(\eta^{1+1/(\beta-1)}), & \beta > 2, \\ O(\eta^2 \log(|s_\mathbf{t}|/\eta)), & \beta = 2. \end{cases}$$

By collecting together the three cases and using Lemma 8 to bound $\rho_\mathbf{t}$, we finish the proof. $\qquad\square$

Using the crude bound $|s_{t_0}| \lesssim 1$ from Lemma 6, it yields

$$\frac{2}{\eta} \geq y_\infty^2 \geq \frac{2}{\eta} - O(\eta) - \begin{cases} O(\eta^{1/(\beta-1)}), & \beta > 2, \\ O(\eta \log(1/\eta)), & \beta = 2, \\ O(\eta), & \beta < 2, \end{cases}$$

which is optimal for $\beta > 2$.

## C.5 EoS regime: quasi-static analysis

We supplement Assumption (A2) with a corresponding lower bound on $\ell'(s)/s$:

$$\text{There exists } C > 0 \text{ such that} \qquad \frac{\ell'(s)}{s} \geq 1 - C\,|s|^\beta \qquad \text{for all } s \neq 0. \tag{A3}$$

Under these assumptions, we prove the following result which is also of interest as it provides detailed information for the bouncing phase of the EoS.

**Theorem 5** (quasi-static principle). *Suppose we run GD on $f$ with step size $\eta > 0$, where $f(x, y) := \ell(xy)$ and $\ell$ satisfies Assumptions (A1), (A2), and (A3). Write $y_t^2 := (2 + \delta_t)/\eta$ and suppose that at some iteration $t_0$, we have $|x_{t_0} y_{t_0}| \asymp \delta_{t_0}^{1/\beta}$ and $\delta_{t_0} \lesssim 1$. Then, for all $t \geq t_0$ with $\delta_t \gtrsim \eta^{\beta/(\beta-1)}$, we have*

$$|x_t y_t| \asymp \delta_t^{1/\beta},$$

*where all implied constants depend on $\ell$ but not on $\eta$.*

In this section, we show that the GD iterates lie close to the quasi-static trajectory and give the full proof of Theorem 2. Recall from (13) that the quasi-static analysis predicts

$$\eta r_t y_t^2 \approx 2 \,, \tag{18}$$

and that during the bouncing phase, this closely agrees with experimental observations (Figure 9). We consider the phase where $y_t^2$ has not yet crossed the threshold $2/\eta$ and we write $y_t^2 := (2 + \delta_t)/\eta$, thinking of $\delta_t$ as small. Then, (18) can be written $(2 + \delta_t)\, r_t \approx 2$. If we have the behavior $\ell'(s)/s = 1 - \Theta(|s_t|^\beta)$ near the origin, then $r_t \approx 1 - \Theta(\delta_t)$ implies that

$$|s_t|^\beta \approx \delta_t \,. \tag{19}$$

Our goal is to rigorously establish (19). However, we first make two observations. First, in order to establish Theorem 2, we only need to prove an upper bound on $|s_t|$, which only requires Assumption (A2) (to prove a lower bound on $|s_t|$, we need a corresponding lower bound on $\ell'(s)/s$). Second, even if we relax (19) to read $|s_t|^\beta \lesssim \delta_t$, this fails to hold when $\delta_t$ is too small, because the error terms (the deviation of the dynamics from the quasi-static trajectory) begin to dominate. With this in mind, we shall instead prove $|s_t|^\beta \lesssim \delta_t + C'\, \eta^\gamma$, where the added $\eta^\gamma$ handles the error terms and the exponent $\gamma > 0$ emerges from the proof.

**Proposition 3** (quasi-static analysis; upper bound). *For all $t$ such that $0 \le \delta_{t-1} \lesssim 1/(\beta \vee 1)$ (for a sufficiently small implied constant), it holds that*

$$|s_t|^\beta \le C \left(\delta_t + C'\, \eta^{\beta/(\beta-1)}\right),$$

*where $C, C' > 0$ are constants which may depend on the problem parameters but not on $\eta$.*

We first show that Theorem 2 now follows.

*Proof of Theorem 2.* As previously noted, the $\beta = +\infty$ case is handled by the arguments of the previous section, so we focus on $\beta < +\infty$. From Lemma 7, we either have $y_t^2 \to 2/\eta$ and $|x_t| \to 0$, in which case we are done, or there is an iteration $\mathbf{t}$ such that $y_{\mathbf{t}}^2 < 2/\eta \le y_{\mathbf{t}-1}^2$. From Proposition 3, since $\delta_{t-1} \ge 0$ and $\delta_t \le 0$, it follows that $|s_{\mathbf{t}}|^\beta \lesssim \eta^{1/(\beta-1)}$. The theorem now follows, either from Proposition 1 or from the refined Proposition 2. $\qquad\square$

We now prove Proposition 3. In the proof, we use asymptotic notation $O(\cdot)$, $\lesssim$, etc. in order to hide constants that depend on $\ell$ (including $\beta$), but not on $\delta_t$ and $\eta$. However, the proof also involves choosing parameters $C, C' > 0$, and we keep the dependence on these parameters explicit for clarity.

*Proof of Proposition 3.* The proof goes by induction; namely, if $|s_t|^\beta \le C\left(\delta_t + C'\eta^\gamma\right)$ and $\delta_t \ge 0$ at some iteration $t$, we prove that the same holds one iteration later, where the constants $C, C' > 0$ as well as the exponent $\gamma > 0$ are chosen later in the proof.

For the base case, observe that the approximate conservation lemma (Lemma 6) gives $|s_t| \lesssim 1$, and $\delta_t \gtrsim 1/(\beta \vee 1)$ at the beginning of the induction, so the bound is satisfied initially if we choose $C$ sufficiently large enough.

Throughout, we also write $\hat{\delta}_t := \delta_t + C'\eta^\gamma$ as a convenient shorthand. The strategy is to prove the following two statements:

1. If $|s_t|^\beta = C_t \hat{\delta}_t$ for some $C_t > \frac{C}{2}$, then $|s_{t+1}|^\beta \le C_{t+1} \hat{\delta}_{t+1}$ for some $C_{t+1} \le C_t$.
2. If $|s_t|^\beta = C_t \hat{\delta}_t$ for some $C_t \le \frac{C}{2}$, then $|s_{t+1}|^\beta \le C \hat{\delta}_{t+1}$.

**_Proof of 1._** The dynamics for $x$ give

$$|x_{t+1}| = |1 - \eta y_t^2 r_t|\, |x_t| \,.$$

By Assumption (A2$^+$) and $|s_t| \lesssim 1$,

$$r_t \le 1 - \Omega(|s_t|^\beta) = 1 - \Omega(C\hat{\delta}_t)$$

and hence

$$\eta y_t^2 r_t = (2 + \delta_t)\left(1 - \Omega(C\hat{\delta}_t)\right) = 2 - \Omega(C\hat{\delta}_t)$$

for large $C$. Also, $\ell''(0) = 1$ and a similar argument as in the proof of Theorem 1 yields the reverse inequality $\eta y_t^2 r_t \gtrsim 1$. We conclude that

$$|x_{t+1}| = \left(1 - \Omega(C\hat{\delta}_t)\right)|x_t|$$

and hence

$$|s_{t+1}|^\beta \le \left(1 - \Omega(C\hat{\delta}_t)\right)|s_t|^\beta = C_t\left(1 - \Omega(C\hat{\delta}_t)\right)\hat{\delta}_t\,.$$

Since we need a bound in terms of $\hat{\delta}_{t+1}$, we use the dynamics of $y$,

$$y_{t+1} = \left(1 - \eta x_t^2 r_t\right)y_t \ge \left(1 - \eta x_t^2\right)y_t\,,$$
$$y_{t+1}^2 \ge \left(1 - 2\eta x_t^2\right)y_t^2\,,$$
$$\delta_{t+1} = \eta y_{t+1}^2 - 2 \ge \delta_t - 2\eta^2 s_t^2 \ge \delta_t - 2\eta^2\left(C\hat{\delta}_t\right)^{2/\beta}\,. \tag{20}$$

Substituting this in,

$$|s_{t+1}|^\beta \le C_t\left(1 - \Omega(C\hat{\delta}_t)\right)\left(\hat{\delta}_{t+1} + 2\eta^2\left(C\hat{\delta}_t\right)^{2/\beta}\right)$$
$$= C_t\hat{\delta}_{t+1} - \Omega(C^2\hat{\delta}_t\hat{\delta}_{t+1}) + 2C\eta^2\left(C\hat{\delta}_t\right)^{2/\beta}\,. \tag{21}$$

Let us show that

$$\hat{\delta}_{t+1} \ge \frac{3}{4}\hat{\delta}_t\,. \tag{22}$$

From (20), we have $\hat{\delta}_{t+1} \ge \hat{\delta}_t - 2\eta^2\left(C\hat{\delta}_t\right)^{2/\beta}$, so we want to prove that $\eta^2\left(C\hat{\delta}_t\right)^{2/\beta} \le \hat{\delta}_t/8$. If $\beta \le 2$ this is obvious by taking $\eta$ small, and if $\beta > 2$ then this is equivalent to $C^{2/\beta}\eta^2 \lesssim \hat{\delta}_t^{1-2/\beta}$. It suffices to have $C^{2/\beta}\eta^2 \lesssim (C')^{1-2/\beta}\eta^{\gamma(1-2/\beta)}$, which is achieved by taking $C'$ large relative to $C$ and by taking $\gamma \le 2/(1 - 2/\beta)$; this constraint on $\gamma$ will be satisfied by our eventual choice of $\gamma = \beta/(\beta - 1)$.

Returning to (21), in order to finish the proof and in light of (22), we want to show that $C^2\hat{\delta}_t^2 \gtrsim C^{1+2/\beta}\eta^2\hat{\delta}_t^{2/\beta}$. Rearranging, it suffices to have $\hat{\delta}_t^{2-2/\beta} \gtrsim C^{2/\beta-1}\eta^2$, or $\hat{\delta}_t^{1-1/\beta} \gtrsim C^{1/\beta-1/2}\eta$. Since by definition $\hat{\delta}_t \ge C'\eta^\gamma$, by choosing $C'$ large it suffices to have $\gamma \le 1/(1 - 1/\beta) = \beta/(\beta-1)$, which leads to our choice of $\gamma$.

***Proof of 2.*** Using the simple bound $\eta y_t^2 r_t \le 2 + \delta_t$, we have

$$|s_{t+1}| \le (1 + \delta_t)|s_t|\,,$$
$$|s_{t+1}|^\beta \le \exp(\beta\delta_t)|s_t|^\beta = C_t\exp(\beta\delta_t)\hat{\delta}_t \le \frac{4}{3}C_t\exp(\beta\delta_t)\hat{\delta}_{t+1}$$

where we used (22). If $\exp(\beta\delta_t) \le 4/3$, which holds if $\delta_t \lesssim 1/\beta$, then from $C_t \le C/2$ we obtain $|s_{t+1}|^\beta \le C\hat{\delta}_{t+1}$ as desired. $\qquad\square$

By following the same proof outline but reversing the inequalities, we can also show a corresponding lower bound on $|s_t|^\beta$, as long as $\delta_t \gtrsim \eta^{\beta/(\beta-1)}$. Although this is not needed to establish Theorem 2, it is of interest in its own right, as it shows (together with Proposition 3) that the iterates of GD do in fact track the quasi-static trajectory.

**Proposition 4** (quasi-static analysis; lower bound)**.** *Suppose additionally that* (A3) *holds and that $\beta < +\infty$. Also, suppose that at some iteration $t_0$, we have $\delta_{t_0} \lesssim 1$ and that*

$$|s_t| \ge c\,\delta_t^{1/\beta} \tag{23}$$

*holds at iteration $t = t_0$, where $c$ is a sufficiently small constant (depending on the problem parameters but not on $\eta$). Then,* (23) *also holds for all iterations $t \ge t_0$ such that $\delta_t \gtrsim \eta^{\beta/(\beta-1)}$.*

*Proof.* The proof mirrors that of Proposition 3. Let $\delta_t \gtrsim \eta^{\beta/(\beta-1)}$ for a sufficiently large implied constant. We prove the following two statements:

1. If $|s_t| = c_t \delta_t^{1/\beta}$ for some $c_t < 2c$, then $|s_{t+1}| \geq c_{t+1} \delta_{t+1}^{1/\beta}$ for some $c_{t+1} \geq c_t$.
2. If $|s_t| = c_t \delta_t^{1/\beta}$ for some $c_t \geq 2c$, then $|s_{t+1}| \geq c \delta_{t+1}^{1/\beta}$.

Throughout the proof, due to Proposition 3, we also have $|s_t| \lesssim \delta_t^{1/\beta}$.

***Proof of 1.*** The dynamics for $x$ give

$$|x_{t+1}| = |1 - \eta y_t^2 r_t| \, |x_t| \, .$$

By Assumption (A3),

$$r_t \geq 1 - O(|s_t|^\beta) \geq 1 - O(c \, \delta_t) \, .$$

If $c$ is sufficiently small, then

$$\eta y_t^2 r_t \geq (2 + \delta_t) \left(1 - O(c \, \delta_t)\right) \geq 2 + \Omega(\delta_t) \, .$$

Therefore, we obtain

$$|x_{t+1}| \geq \left(1 + \Omega(\delta_t)\right) |x_t| \, .$$

On the other hand,

$$y_{t+1} \geq (1 - \eta x_t^2) \, y_t \geq \left(1 - O(\eta^2 s_t^2)\right) y_t \geq \left(1 - O(\eta^2 \delta_t^{2/\beta})\right) y_t \tag{24}$$

and hence

$$|s_{t+1}| \geq \left(1 + \Omega(\delta_t)\right) \left(1 - O(\eta^2 \delta_t^{2/\beta})\right) |s_t| \geq c_t \left(1 + \Omega(\delta_t) - O(\eta^2 \delta_t^{2/\beta})\right) \delta_t^{1/\beta}$$
$$\geq c_t \left(1 + \Omega(\delta_t) - O(\eta^2 \delta_t^{2/\beta})\right) \delta_{t+1}^{1/\beta} \, .$$

To conclude, we must prove that $\eta^2 \delta_t^{2/\beta} \lesssim \delta_t$, but since $\delta_t \gtrsim \eta^{\beta/(\beta-1)}$ (with sufficiently large implied constant), then this holds, as was checked in the proof of Proposition 3.

***Proof of 2.*** Using Assumption (A3),

$$1 - O(\delta_t) \leq 1 - O(|s_t|^\beta) \leq r_t \leq 1 \, .$$

Therefore,

$$2 - O(\delta_t) \leq (2 + \delta_t) \left(1 - O(\delta_t)\right) \leq \eta y_t^2 r_t \leq 2 + \delta_t$$

and

$$-1 + O(\delta_t) \geq 1 - \eta y_t^2 r_t \geq -1 - \delta_t \, .$$

Together with the dynamics for $x$ and (24),

$$|s_{t+1}| \geq \left(1 - O(\delta_t)\right) \left(1 - O(\eta^2 \delta_t^{2/\beta})\right) |s_t| \geq c_t \left(1 - O(\delta_t)\right) \left(1 - O(\eta^2 \delta_t^{2/\beta})\right) \delta_{t+1}^{1/\beta} \, .$$

Since $c_t \geq 2c$, if $\delta_t$ and $\eta$ are sufficiently small it implies $|s_{t+1}| \geq c \, \delta_{t+1}^{1/\beta}$. $\qquad\square$

***Convergence rate estimates.*** Our analysis also provides estimates for the convergence rate of GD in both regimes. Namely, in the gradient flow regime, we show that GD converges in $O(1/\eta)$ iterations, whereas in the EoS regime, GD typically spends $\Omega(1/\eta^{(\beta/(\beta-1))\vee 2})$ iterations ($\Omega(\log(1/\eta)/\eta^2)$ iterations when $\beta = 2$) in the bouncing phase (Figure 11). Hence, the existence of the bouncing phase dramatically slows down the convergence of GD.

*Remark* 2. Suppose that at iteration $t_0$, we have $\delta_{t_0} \asymp 1$. Then, the assumption of Proposition 4 is that $|s_{t_0}| \gtrsim 1$. If this is not satisfied, i.e., $|s_{t_0}| \ll 1$, then the first claim in the proof of Proposition 4 shows that $|s_{t_0+1}| \geq (1 + \Omega(\delta_t)) |s_{t_0}| = (1 + \Omega(1)) |s_{t_0}|$. Therefore, after $t' = O(\log(1/|s_{t_0}|))$ iterations, we obtain $|s_{t_0+t'}| \gtrsim 1$ and then Proposition 4 applies thereafter.

*Remark* 3. From the quasi-static analysis, we can also derive bounds on the length of the bouncing phase. Namely, suppose that $t_0$ is such that $\delta_{t_0} \asymp 1$ and for all $t \geq t_0$, we have $|s_t| = \delta_t^{1/\beta}$. If $\delta_{t_0}$ is sufficiently small so that $r_t \gtrsim 1$ for all $t \geq t_0$, then the equation for $y$ yields

$$\delta_{t+1} \leq \delta_t - \Theta(\eta^2 s_t^2) = \delta_t - \Theta(\eta^2 \delta_t^{2/\beta}).$$

We declare the $k$-th phase to consist of iterations $t$ such that $2^{-k} \leq \delta_t \leq 2^{-(k-1)}$. During this phase, $\delta_{t+1} \leq \delta_t - \Theta(\eta^2 \, 2^{-2k/\beta})$, so the number of iterations in phase $k$ is $\asymp 2^{k \, (2/\beta-1)}/\eta^2$. We sum over the phases until $\delta_t \asymp \eta^{\beta/(\beta-1)}$, since after this point the quasi-static analysis fails and $y_t^2$ crosses over $2/\eta$ shortly afterwards. This yields

$$\frac{1}{\eta^2} \sum_{\substack{k \in \mathbb{Z} \\ \eta^{\beta/(\beta-1)} \lesssim 2^{-k} \lesssim 1}} 2^{k \, (2/\beta-1)} \asymp \begin{cases} 1/\eta^2, & \beta > 2, \\ \log(1/\eta)/\eta^2, & \beta = 2, \\ 1/\eta^{\beta/(\beta-1)}, & \beta < 2. \end{cases}$$

The time spent in the bouncing phase increases dramatically as $\beta \searrow 1$.

# D    Deferred derivations of mean model

In this section, we provide the details for the derivations of the mean model in Subsection 3.1. Recall

$$f(\boldsymbol{x}; a^-, a^+, b) = a^- \sum_{i=1}^d \text{ReLU}\big(-\boldsymbol{x}[i] + b\big) + a^+ \sum_{i=1}^d \text{ReLU}\big(+\boldsymbol{x}[i] + b\big),$$

where $\boldsymbol{x} = \lambda y \boldsymbol{e}_j + \boldsymbol{\xi}$. We first approximate

$$\sum_{i=1}^d \text{ReLU}\big(\pm\boldsymbol{x}[i] + b\big) \approx \sum_{i=1}^d \text{ReLU}\big(\pm\boldsymbol{\xi}[i] + b\big).$$

In other words, we can ignore the contribution of the signal $\lambda y \boldsymbol{e}_j$. This approximation holds because *(i)* initially, the bias $b$ is not yet negative enough to threshold out the noise, and hence the summation $\sum_{i=1}^d \text{ReLU}(\pm\boldsymbol{\xi}[i] + b)$ is of size $O(d)$, and *(ii)* the difference between the left- and right-hand sides above is simply $\text{ReLU}(\pm\lambda y \pm \boldsymbol{\xi}[j] + b) - \text{ReLU}(\pm\boldsymbol{\xi}[j] + b)$, which is of size $O(1)$ and hence negligible compared to the full summation.

Next, letting $g(b) := \mathbb{E}_{z \sim \mathcal{N}(0,1)} \text{ReLU}(z + b)$ be the 'smoothed' ReLU (see Figure 7), concentration of measure implies the following two facts:

- $\sum_{i=1}^d \text{ReLU}\big(\pm\boldsymbol{\xi}[i] + b\big) \approx d\,\mathbb{E}_{\xi \sim \mathcal{N}(0,1)} \text{ReLU}(\xi + b) =: d\,g(b)$ and
- $\sum_{i=1}^d \mathbb{1}\{\pm\boldsymbol{x}[i] + b \geq 0\} \approx d\,\mathbb{E}_{\xi \sim \mathcal{N}(0,1)} \mathbb{1}\{\xi + b \geq 0\} = d\,g'(b).$

Indeed, the summations above are sums of $d$ i.i.d. non-negative random variables, and hence its mean is $\Omega(d)$ (as long as $b \geq -O(1)$) and its standard deviation is $O(\sqrt{d})$. Now, using these approximations, one can rewrite the GD dynamics on the population loss $\mathbb{E}[\ell_{\text{logi}}(y f(\boldsymbol{x}; a^-, a^+, b))]$.

Using these approximations, the output of the ReLU network (2) can be written as

$$f(\boldsymbol{x}; a^-, a^+, b) \approx d\,(a^- + a^+)\,g(b),$$

which in turn leads to an approximation of the GD dynamics on the population loss $(a^-, a^+, b) \mapsto \mathbb{E}[\ell_{\text{logi}}(y f(\boldsymbol{x}; a^-, a^+, b))]$:

$$a_{t+1}^\pm = a_t^\pm - \eta\,\mathbb{E}\Big[\ell'_{\text{logi}}\big(y \underbrace{f(\boldsymbol{x}; a_t^-, a_t^+, b_t)}_{\approx d\,(a_t^- + a_t^+)\,g(b_t)}\big) \times \underbrace{\sum_{i=1}^d \text{ReLU}\big(\pm\boldsymbol{x}[i] + b_t\big)}_{\approx d\,g(b_t)}\Big]$$

$$\approx a_t^\pm - \eta\,\ell'_{\text{sym}}\big(d\,(a_t^- + a_t^+)\,g(b_t)\big)\,d\,g(b_t),$$

$$b_{t+1} = b_t - \eta\,\mathbb{E}\Big[\ell'_{\text{logi}}\big(y \underbrace{f(\boldsymbol{x}; a_t^-, a_t^+, b_t)}_{\approx d\,(a_t^- + a_t^+)\,g(b_t)}\big) \times \Big(a_t^- \underbrace{\sum_{i=1}^d \mathbb{1}\{-\boldsymbol{x}[i] + b_t \geq 0\}}_{\approx d\,g'(b_t)} + a_t^+ \underbrace{\sum_{i=1}^d \mathbb{1}\{+\boldsymbol{x}[i] + b_t \geq 0\}}_{\approx d\,g'(b_t)}\Big)\Big]$$

$$\approx b_t - \eta\,\ell'_{\text{sym}}\big(d\,(a_t^- + a_t^+)\,g(b_t)\big)\,d\,(a_t^- + a_t^+)\,g'(b_t),$$

where $\ell_{\mathrm{sym}}(s) := \frac{1}{2}(\log(1 + \exp(-s)) + \log(1 + \exp(+s)))$ is the symmetrized logistic loss. Hence we arrive at the following dynamics on $a^{\pm}$ and $b$ that we call the mean model:

$$a_{t+1}^{\pm} = a_t^{\pm} - \eta\,\ell'_{\mathrm{sym}}\big(d\,(a_t^- + a_t^+)\,g(b_t)\big)\,d\,g(b_t)\,,$$
$$b_{t+1} = b_t - \eta\,\ell'_{\mathrm{sym}}\big(d\,(a_t^- + a_t^+)\,g(b_t)\big)\,d\,(a_t^+ + a_t^-)\,g'(b_t)\,.$$

Now, we can write the above dynamics more compactly in terms of the parameter $A_t := d\,(a_t^- + a_t^+)$.

$$A_{t+1} = A_t - 2d^2\eta\,\ell'_{\mathrm{sym}}(A_t g(b_t))\,g(b_t)\,,$$
$$b_{t+1} = b_t - \eta\,\ell'_{\mathrm{sym}}(A_t g(b_t))\,A_t g'(b_t)\,.$$

# E    Proofs for the mean model

In this section, we prove the main theorems for the mean model. We first recall the mean model for the reader's convenience.

$$A_{t+1} = A_t - 2d^2\eta\,\ell'(A_t g(b_t))\,g(b_t)\,,$$
$$b_{t+1} = b_t - \eta\,\ell'(A_t g(b_t))\,A_t g'(b_t)\,.$$

## E.1    Deferred proofs

In this section, we collect together deferred proofs from Subsection 3.2.

*Proof of Lemma 2.* By definition, $g(b) = \int_{-b}^{\infty}(\xi + b)\,\varphi(\xi)\,\mathrm{d}\xi = \int_{-b}^{\infty}\xi\,\varphi(\xi)\,\mathrm{d}\xi + b\,\Phi(b)$. Recalling $\varphi'(\xi) = -\xi\,\varphi(\xi)$, the first term equals $\varphi(b)$. Moreover, $g'(b) = -b\,\varphi(b) + \Phi(b) + b\,\varphi(b) = \Phi(b)$.    □

*Proof of Lemma 3.* Note that $\partial_t(\frac{1}{2}A^2) = A\dot{A} = -2d^2\,\ell'(Ag(b))\,Ag(b)$ and also that $\partial_t\kappa(b) = -\ell'(Ag(b))\,\kappa'(b)\,Ag'(b) = -\ell'(Ag(b))\,Ag(b)$ since $\kappa' = g/g'$. Hence, $\partial_t\big(\frac{1}{2}A^2 - 2d^2\kappa(b)\big) = 0$ and the proof is completed.    □

## E.2    Gradient flow regime

*Proof of Theorem 3.* The following proof is analogous to the proof of Theorem 1. We first list several facts we use in the proof:

(i)  $|g'(b)| = |\Phi(b)| \le 1$ for all $b \in \mathbb{R}$.

(ii)  $\ell'(s) = \frac{1}{2}\frac{\exp(s)-1}{\exp(s)+1}$. Hence, $|\ell'(s)| \le \frac{1}{2}$ for all $s \in \mathbb{R}$, and we have

$$\frac{\ell'(s)}{s} \ge \frac{1}{8} \times \begin{cases} 1\,, & \text{if } |s| \le 2\,, \\ 2/|s|\,, & \text{if } |s| > 2\,. \end{cases}$$

(iii)  $\ell''(0) = 1/4$.

(iv)  $\ell'''(s) = -\frac{\exp(s)\,(\exp(s)-1)}{(\exp(s)+1)^3}$. Hence, $\ell'''(s) < 0$ for $s > 0$ and $\ell'''(s) > 0$ for $s < 0$. In particular, $|\ell'(s)| \le \frac{1}{4}|s|$ for all $s \in \mathbb{R}$.

Throughout the proof, we assume that $A_0 > 0$ without loss of generality. We prove by induction the following claim: for $t \ge 0$ and

$$\gamma := \frac{1}{200}\min\Big\{\delta,\, 8 - \delta,\, \frac{8 - \delta}{A_0}\Big\}\,,$$

it holds that $|A_t| \le A_0\exp(-\gamma t)$. This clearly holds at initialization.

Suppose that the claim holds up to iteration $t$. Using the bounds on $|g'|$ and $|\ell'|$, it follows that

$$b_{t+1} \ge b_t - |\ell'(A_t g(b_t))|\,|A_t|\,g'(b_t) \ge b_t - \frac{1}{2}\eta\,|A_t|$$

$$\ge b_t - \frac{1}{2}\eta A_0\exp(-\gamma t) \ge \cdots \ge b_0 - \frac{1}{2}\eta A_0\sum_{s=0}^{t}\exp(-\gamma s) \ge -\frac{\eta A_0}{\gamma}\,.$$

In particular, $b_t \geq -1$ and $g(b_t) > 0.08$, since $\eta \leq \frac{\gamma}{A_0}$. Also, the bound shows that if the claim holds for all $t$, then we obtain the desired conclusion.

It remains to establish the inductive claim; assume that it holds up to iteration $t$. For the dynamics of $A$, by symmetry we may suppose that $A_t > 0$. From $\ell'(A_t g(b_t)) \leq A_t g(b_t)/4$ and $g(b_t) \leq g(0) = \frac{1}{\sqrt{2\pi}}$, it follows that

$$
\begin{aligned}
A_{t+1} = A_t - 2\eta d^2 \, \ell'(A_t g(b_t)) \, g(b_t) &\geq \left(1 - \frac{\eta d^2}{2} g(b_t)^2\right) A_t \\
&\geq \left(1 - \frac{\eta d^2}{2} g(0)^2\right) A_t = -\left(1 - \frac{\delta}{4}\right) A_t \, .
\end{aligned}
$$

This shows that $A_{t+1} \geq -(1 - \gamma) A_t$. Next, we show that $A_{t+1} \leq (1 - \gamma) A_t$. First, if $A_t g(b_t) \leq 2$,

$$
\begin{aligned}
A_{t+1} = A_t - 2\eta d^2 \, \ell'(A_t g(b_t)) \, g(b_t) &\leq A_t - \frac{1}{4} \eta d^2 A_t \, g(b_t)^2 \\
= \left(1 - \frac{(8 - \delta)\pi}{4} g(b_t)^2\right) A_t &\leq \left(1 - \frac{(8 - \delta)}{4}\pi \cdot 0.08^2\right) A_t \leq (1 - \gamma) A_t \, ,
\end{aligned}
$$

since we have $g(b_t) > 0.08$. Next, if $A_t g(b_t) \geq 2$, then

$$
\begin{aligned}
A_{t+1} = A_t - 2\eta d^2 \, \ell'(A_t g(b_t)) \, g(b_t) &\leq A_t - \frac{1}{2} \eta d^2 g(b_t) = \left(1 - \frac{(8 - \delta)\pi}{2} \frac{g(b_t)}{A_t}\right) A_t \\
\leq \left(1 - \frac{(8 - \delta)\pi}{2} \cdot \frac{0.08}{A_0}\right) A_t &\leq (1 - \gamma) A_t \, .
\end{aligned}
$$

This shows that $|A_{t+1}| \leq (1 - \gamma)|A_t|$ for the case $A_t > 0$. A similar conclusion is obtained for the case $A_t < 0$. The induction is complete. □

### E.3 EoS regime

*Proof of Theorem 4.* The following proof is analogous to the proof of Lemma 7. Assume throughout that $A_t \neq 0$ for all $t$. Recall the dynamics for $b$:

$$
b_{t+1} = b_t - \eta \, \ell'(A_t g(b_t)) \, A_t g'(b_t) \, .
$$

Since $\ell'(s)/s \to 1/4$ as $s \to 0$, and $\ell'$ is increasing, this equation implies that if $\liminf_{t\to\infty} |A_t| > 0$ then $b_t$ must keep decreasing until $\frac{1}{2} d^2 g(b_t)^2 < 2/\eta$.

Suppose for the sake of contradiction that there exists $\varepsilon > 0$ with $\frac{1}{2} d^2 g(b_t)^2 > (2 + \varepsilon)/\eta$, for all $t$. Let $\varepsilon' > 0$ be such that $1 - (2 + \varepsilon)(1 - \varepsilon') < -1$, i.e., $\varepsilon' < \frac{\varepsilon}{2+\varepsilon}$. Then, there exists $\delta > 0$ such that $|A_t| \leq \delta$ implies $\ell'(A_t g(b_t))/(A_t g(b_t)) > \frac{1}{4}(1 - \varepsilon')$, hence

$$
\frac{|A_{t+1}|}{|A_t|} = \left|1 - 4 \cdot \frac{1}{4}(1 - \varepsilon') \cdot \frac{1}{2} \eta d^2 g(b_t)^2\right| > |(2 + \varepsilon)(1 - \varepsilon') - 1| > 1 \, .
$$

The above means that $|A_t|$ increases until it exceeds $\delta$, i.e., $\liminf_{t\to\infty} |A_t| \geq \delta$. This is our desired contradiction and it implies that $\lim_{t\to\infty} \frac{1}{2} d^2 g(b_t)^2 \leq 2/\eta$. □

*Remark 4.* A straightforward calculation yields that when $(a_\star^-, a_\star^+, b_\star)$ is a global minimizer (i.e., $a_\star^- + a_\star^+ = 0$), then $\lambda_{\max}(\nabla^2 f(a_\star^-, a_\star^+, b_\star)) = \frac{1}{2} d^2 g(b_\star)^2$. The mean model initialized at $(A_0, 0)$ approximately reaches $(0, 0)$ whose sharpness is $d^2 g(0)^2/2 = d^2/4\pi$. Hence, the bias learning regime $2/\eta < d^2/(4\pi)$ precisely corresponds to the EoS regime, $2/\eta < \lambda_{\max}(\nabla^2 f(a_\star^-, a_\star^+, b_\star))$.

