$$\geq \Big(1 - \frac{\eta d^2}{2}\, g(0)^2\Big) A_t = -\Big(1 - \frac{\delta}{4}\Big) A_t.$$

This shows that $A_{t+1} \geq -(1-\gamma)\, A_t$. Next, we show that $A_{t+1} \leq (1-\gamma)\, A_t$. First, if $A_t g(b_t) \leq 2$,

$$A_{t+1} = A_t - 2\eta d^2\, \ell'(A_t g(b_t))\, g(b_t) \leq A_t - \frac{1}{4}\eta d^2\, A_t\, g(b_t)^2$$

$$= \Big(1 - \frac{(8-\delta)\, \pi}{4}\, g(b_t)^2\Big) A_t \leq \Big(1 - \frac{(8-\delta)}{4}\, \pi \cdot 0.08^2\Big) A_t \leq (1-\gamma)\, A_t,$$

since we have $g(b_t) > 0.08$. Next, if $A_t g(b_t) \geq 2$, then

$$A_{t+1} = A_t - 2\eta d^2 \ell'(A_t g(b_t)) g(b_t) \leq A_t - \frac{1}{2}\eta d^2 g(b_t) = \left(1 - \frac{(8-\delta)\pi}{2} \frac{g(b_t)}{A_t}\right) A_t$$

$$\leq \left(1 - \frac{(8-\delta)\pi}{2} \cdot \frac{0.08}{A_0}\right) A_t \leq (1-\gamma) A_t.$$

This shows that $|A_{t+1}| \leq (1-\gamma)|A_t|$ for the case $A_t > 0$. A similar conclusion is obtained for the case $A_t < 0$. The induction is complete. $\qquad\square$

### D.3 EoS regime

*Proof of Theorem 4.* The following proof is analogous to the proof of Lemma 7. Assume throughout that $A_t \neq 0$ for all $t$. Recall the dynamics for $b$:

$$b_{t+1} = b_t - \eta\, \ell'(A_t g(b_t)) A_t g'(b_t).$$

Since $\ell'(s)/s \to 1/4$ as $s \to 0$, and $\ell'$ is increasing, this equation implies that if $\liminf_{t\to\infty} |A_t| > 0$ then $b_t$ must keep decreasing until $\frac{1}{2} d^2 g(b_t)^2 < 2/\eta$.

Suppose for the sake of contradiction that there exists $\varepsilon > 0$ with $\frac{1}{2} d^2 g(b_t)^2 > (2+\varepsilon)/\eta$, for all $t$. Let $\varepsilon' > 0$ be such that $1 - (2+\varepsilon)(1-\varepsilon') < -1$, i.e., $\varepsilon' < \frac{\varepsilon}{2+\varepsilon}$. Then, there exists $\delta > 0$ such that $|A_t| \leq \delta$ implies $\ell'(A_t g(b_t))/(A_t g(b_t)) > \frac{1}{4}(1-\varepsilon')$, hence

$$\frac{|A_{t+1}|}{|A_t|} = \left|1 - 4 \cdot \frac{1}{4}(1-\varepsilon') \cdot \frac{1}{2}\eta d^2 g(b_t)^2\right| > |(2+\varepsilon)(1-\varepsilon') - 1| > 1.$$

The above means that $|A_t|$ increases until it exceeds $\delta$, i.e., $\liminf_{t\to\infty} |A_t| \geq \delta$. This is our desired contradiction and it implies that $\lim_{t\to\infty} \frac{1}{2} d^2 g(b_t)^2 \leq 2/\eta$. $\qquad\square$

*Remark* 4. A straightforward calculation yields that when $(a_\star^-, a_\star^+, b_\star)$ is a global minimizer (i.e., $a_\star^- + a_\star^+ = 0$), then $\lambda_{\max}(\nabla^2 f(a_\star^-, a_\star^+, b_\star)) = \frac{1}{2} d^2 g(b_\star)^2$. The mean model initialized at $(A_0, 0)$ approximately reaches $(0,0)$ whose sharpness is $d^2 g(0)^2/2 = d^2/4\pi$. Hence, the bias learning regime $2/\eta < d^2/(4\pi)$ precisely corresponds to the EoS regime, $2/\eta < \lambda_{\max}(\nabla^2 f(a_\star^-, a_\star^+, b_\star))$.

## E  Additional figures

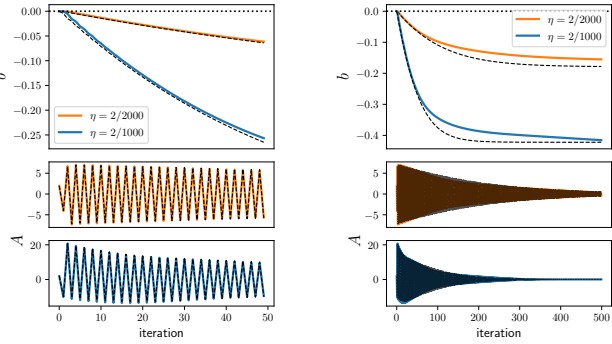

Figure 10: Under the same setting as Figure 1, we compare the mean model with the GD dynamics of the ReLU network. The mean model is plotted with black dashed line. Note that **the mean model tracks the GD dynamics quite well during the initial phase of training.**

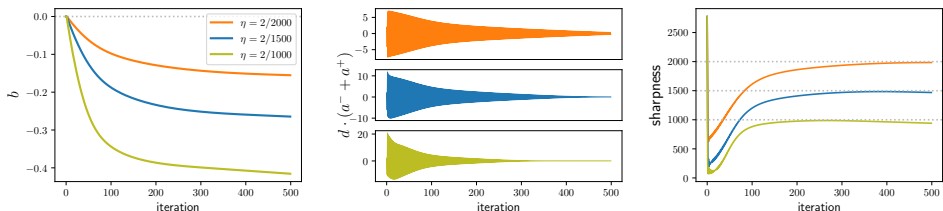

Figure 11: **Understanding our main question is surprisingly related to the EoS.** Under the same setting as Figure 1, we report the largest eigenvalue of the Hessian ("sharpness"), and observe that GD iterates lie in the EoS during the initial phase of training when there is a fast drop in the bias.

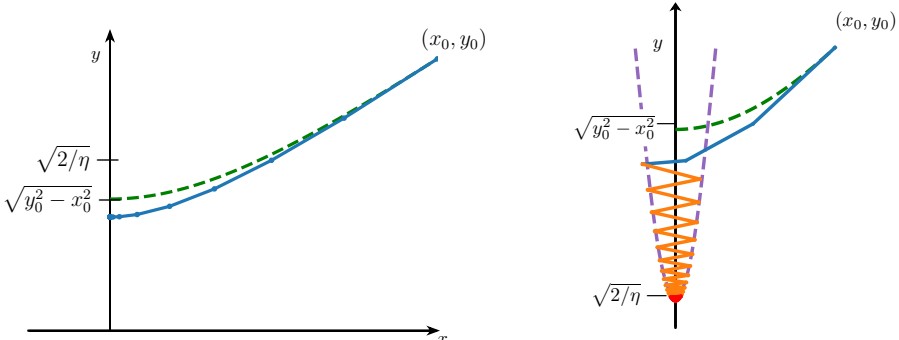

Figure 12: **Two regimes for GD.** We run GD on the square root loss with step size $\frac{1}{4}$. The gradient flow regime is illustrated on the left for $(x_0, y_0) = (3, 4)$. GD (blue) tracks the gradient flow (green) when $\eta < 2/(y_0^2 - x_0^2)$. Otherwise, as illustrated on the right for $(x_0, y_0) = (3, 6)$, GD is in the EoS regime and goes through a gradient flow phase (blue), an intermediate bouncing phase (orange) that tracks the quasi-static envelope (purple), and a converging phase (red).