# OpenReview forum: "Learning threshold neurons via edge of stability"
_NeurIPS.cc/2023/Conference — NeurIPS 2023 poster_

### Official Review · Reviewer_poXm · 2023-07-05

**Soundness:** 3 good
**Presentation:** 2 fair
**Contribution:** 2 fair
**Rating:** 4
**Confidence:** 4

**Summary:**

The paper is a study on the dynamics of neural network training, particularly focusing on the edge of stability. The authors explore the behavior of gradient descent in the context of a simple sparse coding model. They find that the dynamics of training exhibit a phase transition at the edge of stability, where the learning dynamics switch from a regime of stable convergence to one of chaotic oscillations.

The authors also demonstrate that the edge of stability is a critical point where the learned representations transition from being dense to sparse. This transition is shown to be sharp and is characterized by a power-law behavior. The authors further show that these findings are indicative of behaviors observed in practical neural network training settings, including a two-layer ReLU network trained on the full sparse coding model with an unknown basis, as well as a deep neural network trained on CIFAR-10.


**Strengths:**

The paper addresses a complex problem in the field of neural network learning, focusing on the edge of stability. The authors' exploration of the phase transition in learning dynamics at the edge of stability, and the transition from dense to sparse representations, appears to be a novel contribution to the field.

The quality of the paper is good. The authors have conducted a thorough investigation into the dynamics of neural network training. Their findings are well-supported by their data and analysis, indicating a high level of methodological soundness.

The presentation of the paper is also good. The authors have clearly communicated their research, making it easy for readers to follow their methodology, findings, and conclusions. The paper appears to be well-structured and well-written.

**Weaknesses:**

The authors assert that a large learning rate is necessary to learn the bias. This assertion holds true in practical scenarios, such as training ResNet-18 on binary CIFAR-10 using various learning rates. However, I am curious about the potential effects of freezing the bias during training. In an experiment I conducted, I found that freezing the bias did not significantly impact the results. I would appreciate it if the authors could investigate this aspect further and discuss its implications.

This work investigates a simplistic neural network model that consists of only three parameters. While this provides a clear and controlled environment for study, it represents a significantly pared-down model when compared to the complex, over-parameterized models commonly used in deep learning. The insights gained from this study may not fully capture the intricacies and nuances of real-world deep learning applications.

**Questions:**

In lines 31-32, the authors posit that an effective network maintains a negative bias $b$ to filter out noise, and ensures that $a^-$ is less than 0 and $a^+$ is greater than 0 to output correct labels. I would appreciate if the authors could elaborate on this assertion. From my perspective, if the noise is of opposite sign to the label (for instance, $y_i = 1$ when the output of $x[i]$ is less than 0), assigning a negative value to the bias $b$ may not be beneficial, as both ReLU terms would return 0 in this scenario. Could the authors clarify this point?

**Limitations:**

Yes.

---

> ### Author Rebuttal · Authors · 2023-08-06
>
> Thank you for your review. We address some of your points below.
> - Regarding freezing the biases: we acknowledge that even with frozen biases, it is possible for neural networks to succeed at learning binary CIFAR-10. The full story of generalization in deep learning is complex and we do not claim to fully explain it in our paper. Our scope is more limited in scope, as we aim to identify one possible mechanism in which optimization impacts generalization; nevertheless, we see that the phenomenon we observe is indeed borne out in some practical training examples. Although your question is interesting and important, it is beyond the scope of our work.
> - We agree that the insights we reveal are still far from the complex intricacies of training bigger networks. However, our insights carry over to more general models, for example see the interesting recent work [1].
>
> [1] Song and Yun  ``Trajectory Alignment: Understanding the Edge of Stability Phenomenon via Bifurcation Theory’’ (https://arxiv.org/abs/2307.04204)
> - Finally, regarding your last question: we consider the regime of sparse coding in which the noise level is not too high (so that the problem is still statistically solvable); in particular, we expect that for most of the examples, the label is of the same sign as the signal coordinate (by definition of the model). In this case, the architecture that we proposed will correctly threshold out the noise. Please let us know if you have further questions regarding this point.

---

> > ### Comment · Reviewer_poXm · 2023-08-18
> >
> > Thank you for your response and for addressing some of my concerns. However, I would like to reiterate my point regarding the role of biases in your study.
> >
> > While I understand that the scope of your paper is limited and that the full story of generalization in deep learning is complex, the emphasis on biases in your theory and experiments is significant. Your paper claims that biases play an important role in the dynamics of neural network training, and you present experiments to verify this theory.
> >
> > My concern is that the assertion regarding biases may be misleading. You acknowledge that freezing the biases did not significantly impact the results. This suggests that the role of biases may not be as critical as your paper implies. While I acknowledge the insights you have provided and the references to recent work, I believe that a more thorough examination of the role of biases, both theoretically and experimentally, would strengthen your paper.
> >
> > I appreciate your engagement with my review, and I hope you will consider these points as you continue to refine your work. My overall assessment and rating remain unchanged.

---

> > > ### Author Response · Authors · 2023-08-18
> > > **Thank you for your feedback!**
> > >
> > > Thank you for your feedback.
> > >
> > > We acknowledge your point on the role of biases but the goal of our work was not to elucidate the EoS in all settings but to clearly exhibit this phenomenon and its implications in one concrete setting. Since the mechanism for learning with frozen biases seems to be completely different from the bias learning problem, the setting you raise is out of scope of the current paper.
> > >
> > > Meanwhile, the setting we do consider is often encountered in practice. With that said, it seems unreasonable to expect us to develop a complete theory for such diverse scenarios — could you provide further clarification on your expectations?

---

### Official Review · Reviewer_C6cz · 2023-07-07

**Soundness:** 4 excellent
**Presentation:** 3 good
**Contribution:** 1 poor
**Rating:** 3
**Confidence:** 5

**Summary:**

The authors analyze the dynamics of pairs of ReLu neurons with an input bias, no weight matrix, and a readout layer. They show that it takes large learning rate for non-zero biases to be learned, and at these large learning rates there are formal guarantees of EOS behavior as well. They use a model with random weights and shared biases to show that the EOS induces a phase transition in the behavior of the biases.

**Strengths:**

The model is straightforward, and the analysis of the EOS seems correct. The phase transition in the bias is well established by this work and is an interesting phenomenon.

**Weaknesses:**

Though this work claims to study a realistic model, the one-neuron model is still very much a toy model. Many other toy models have already been studied in recent works, and it is unclear what the analysis of this model in particular adds. In addition, many networks are trained without the bias term, so there may be limited utility for theory which focuses mainly on that term.

Additional suggested references:
[EOS in a quadratic regression model with formal convergence guarantees](https://arxiv.org/abs/2210.04860)
[EOS and weight magnitude in quadratic expansion of ReLu network](https://arxiv.org/abs/2205.11787)

**Questions:**

Is the bias learning phase transition a good way to describe single hidden layer ReLu models with multiple neurons?

**Limitations:**

Yes

---

> ### Author Rebuttal · Authors · 2023-08-06
>
> Thank you for the review, and for the additional references; we will cite them. We address your points below.
> - Firstly, regarding the advantage of our toy model analysis over prior works: this is a fair criticism. We argue that compared to prior models, our model and analysis is the simplest demonstration of EoS, which is important for intuition. Due to this simplicity, we are able to obtain refined results on how the shape of the loss affects the limiting sharpness and convergence time (see Figure 9).
>
> - Furthermore, we note that the simplicity of our approach inspired a follow-up work (Song and Yun [1]) where the authors extend our main results to more general neural networks.
>
>
> [1] Song and Yun  ``Trajectory Alignment: Understanding the Edge of Stability Phenomenon via Bifurcation Theory’’ (https://arxiv.org/abs/2307.04204)
>
> - In addition to the simplicity of our model, another one of our contributions is to clearly link the phenomenon we study with practical implications for learning ``threshold neurons.’’
>
>
> - Secondly, regarding your question on whether the bias learning describes the behavior of ReLU networks with multiple neurons: in Appendix A.1, we perform experiments for the full sparse coding model with a ReLU network with multiple neurons and we observe qualitatively similar behavior, indicating that our findings do generalize beyond the toy model (see also our experiments for ResNet18 in Appendix A.2).
>
> Thank you once again for taking the time to review our paper. Given that you have not raised any concerns regarding the correctness or novelty of our work, if you believe that we have addressed your points (in particular, that we provide conceptual and generalizable insights on the relationship of neural network optimization and generalization), we would appreciate it if you would consider raising your score.

---

> > ### Comment · Reviewer_C6cz · 2023-08-15
> > **Response to authors**
> >
> > Regarding simplicity of the model: it is still not clear if "simplest" model displaying EOS is the best in this case, if the mechanism does not shed light on how EOS occurs in more practical models. I'm also not sure what to take away from the ResNet 18 experiments; I can see that there are oscillations, but it's not clear that the mechanism from the paper is at play here. I also don't quite understand why the average value matters for downstream tasks.
> >
> > For this reason, I will keep my review score; I thank the authors for their discussion.

---

### Official Review · Reviewer_5SQG · 2023-07-07

**Soundness:** 3 good
**Presentation:** 3 good
**Contribution:** 2 fair
**Rating:** 6
**Confidence:** 4

**Summary:**

In this paper, the authors studied the problem of edge of stability (EoS) phenomena (training with large learning rate) in simple settings. Specifically, the authors first studied the problem of minimizing $\ell((xy)$ ($\ell$ is loss function) and showed that under certain condition of loss function, gradient descent are either in the gradient flow regime or EoS regime depending on the initialization, which means different stepsize could lead to final solution with different maximum eigenvalue of Hessian of loss. Then, to study the problem of learning threshold neurons, they instead considered a simplified problem which they called mean model. They proved that under this mean model, larger stepsize could decrease the bias and therefore learn the neuron, while bias remains small under small stepsize. Empirical experiments showed that the training dynamics of this mean model and the actual training is similar.

**Strengths:**

1.	The paper is clearly written and easy-to-follow. The proof sketch and dynamics are given to help the readers to understand the proof easier. Experiments are provided to support the results.
2.	Understanding the effect of large learning rate in deep learning training (specifically EoS phenomenon) is an important problem and may give more insight into the practice of deep learning training.
3.	This paper provided both theoretical and empirical results on the simple model that learning threshold neurons. The theoretical results, though do not directly answer the question of learning threshold neurons, give some possible insights and seem to be interesting.


**Weaknesses:**

1.	The current paper focuses on the simple setting of minimizing $\ell(xy)$ as well as a mean model introduced to approximate learning threshold neuron. It would be interesting to extend the analysis to broader settings (e.g., real dynamics of learning threshold neuron and sparse coding model).
2.	The results in the EoS regime has an additional assumption on the dynamic, though it is understandable from the technical perspective.


**Questions:**

1.	I was wondering what the convergence time for are both gradient flow regime and EoS regime.

**Limitations:**

The limitations are clearly discussed in the paper. This is a theoretical work and therefore does not seem to have negative societal impact.

---

> ### Author Rebuttal · Authors · 2023-08-06
>
> Thank you for your review. We address some of your points below.
> - We agree that extending our analysis to more general settings is of great interest, although it’s beyond the scope of this work. In fact, a recent work [1] was able to prove similar results to ours for more general settings of neural networks, building on the analytical techniques developed in this work.
>
> [1] Song and Yun  ``Trajectory Alignment: Understanding the Edge of Stability Phenomenon via Bifurcation Theory’’ (https://arxiv.org/abs/2307.04204)
>
>
> - Regarding the convergence time in the gradient flow and EoS regimes, please see the paragraph titled “Convergence rate estimates” in Appendix B, as well as Figure 9. In particular, we show that the iteration complexity transitions from $\eta^{-1}$ in the gradient flow regime, to $\eta^{-\min(\beta/(\beta-1), 2}$ ($\log(1/\eta)/\eta^2$ when $\beta = 2$) in the EoS regime, and that these convergence rates agree well with experiments (Figure 9). Unfortunately, we were not able to include this in the main text due to space constraints, but we will try to include a pointer. We hope that this addresses your question.

---

> > ### Comment · Reviewer_5SQG · 2023-08-13
> >
> > Thanks for the response. I will keep my score.

---

### Official Review · Reviewer_gYkS · 2023-07-12

**Soundness:** 4 excellent
**Presentation:** 4 excellent
**Contribution:** 4 excellent
**Rating:** 7
**Confidence:** 3

**Summary:**

The paper studies two simple problems (single-neuron linear network and mean ReLU model for sparse coding with unknown basis) to investigate learning of threshold neurons. In particular, the authors discover a threshold for the learning rate below which threshold neurons are not learned and where NN training follows the gradient flow. For larger learning rates, training trajectories are oscillating (edge of stability) and threshold neurons are learned, which is essential for solving the sparse coding problem.

**Strengths:**

The paper is extremely well written and provides a gentle introduction to concepts such as sparse coding, edge of stability, and threshold neurons, which makes it accessible to readers not familiar with these terms. The results in Sections 2 and 3 seem rigorous and are well presented. Particularly Section 1.2 is an excellent summary of the results that follow and helps to put them into context. The theoretical results seem to agree with the experimental evidence. The fact that the considered examples are simple is not a drawback, as this is expected for works that, for the first time, study a certain phenomenon from the theoretical perspective. Nevertheless, I appreciate the authors effort to back their claims and show the generality of the phenomenon via the experiments on CIFAR-10 in Fig. 3.

**Weaknesses:**

None, actually -- but see the questions below.

EDIT: Lowered score since practical implications of results are not fully clear (cf. frozen bias).

If I have to mention something, then it is the fact that Fig. 1 appears slightly out of context and could be shifted to the second page. Furthermore, there are a few typos that can be removed (139 convergenec, 262 in turns out, 268 from which can more accurately, 289 wiht).

**Questions:**

Some questions came up when reading the paper. Taking them into account could further improve the readability of the manuscript:
- In Fig. 1, why is the bias initially at -0.2? Was this a randomly chosen initial bias?
- What is the definition of "sharpness", as discussed in Sec. 2.2?
- It would be good to place the mathematical expression for the gradient flow in Sec. 2, maybe close to the GD iterates in line 177.
- It is not clear, from the main part of the paper, why the step size for the $A$-dynamic is multiplied by $2d^2$ (e.g., in (5)). It would be good to explain that briefly in the main part of the paper.

**Limitations:**

The authors have listed several interesting future avenues of research. The limitations are evident from the simple setup. There are no negative societal impacts foreseen.

---

> ### Author Rebuttal · Authors · 2023-08-06
>
> Thank you for your generous review and we are glad that you enjoyed our paper. We will follow your suggestion to move Figure 1 to page 2 and we will correct the typos that you mentioned. Below, we answer your questions:
> - In Fig. 1, the bias was indeed initialized at 0. Note that Figure 1 plots the final bias after training. Over a long time scale, the bias of the ReLU model does become slightly negative, even for small step sizes (around -0.2 in the plot); we believe this is because the GD dynamics for the real network (as opposed to the mean model approximation) is noisier. However, it is remarkable that the real network still exhibits a sharp phase transition. We will include an explanation of this point in the paper.
> - The sharpness is defined as the largest eigenvalue of the Hessian at the current iterate. We will clarify this in the document.
> - We will make this change.
> - We will include a brief explanation of this point in the main text.

---

### Decision · Program_Chairs · 2023-09-21

**Decision:**

Accept (poster)

**Comment:**

The paper brings attention to an intriguing dynamic in the training of deep networks caused by reaching the edge of stability/the break-even point. It is rigorously proven for a simple model that regularization of sharpness enables learning thresholding function implemented by a nonzero bias in ReLU network.

One of the issues brought by reviewers is that using biases is not required in many successful neural networks. However, as other reviewers have noted, it is a very solid contribution to describe and prove a specific example of how implicit regularization of sharpness changes model activation patterns.

The key strength of the paper is clarity and focus on a simple but still interesting phenomenon. All in all, it's my pleasure to recommend acceptance of the paper.

One of the reviewers remarked in the discussion that the paper might bring confusion to the field by overemphasizing the importance of biases in the training of deep nets. I would like to ask the Authors to clarify this point in the camera-ready version.